# Cross-Linked Alginate Dialdehyde/Chitosan Hydrogel Encompassing Curcumin-Loaded Bilosomes for Enhanced Wound Healing Activity

**DOI:** 10.3390/pharmaceutics16010090

**Published:** 2024-01-09

**Authors:** Sarah A. Sideek, Hala B. El-Nassan, Ahmed R. Fares, Nermeen A. Elkasabgy, Aliaa N. ElMeshad

**Affiliations:** 1Department of Pharmaceutics and Industrial Pharmacy, Faculty of Pharmacy, Cairo University, Cairo 11562, Egypt; sarahsideek91@gmail.com (S.A.S.); ahmed.roshdy@pharma.cu.edu.eg (A.R.F.); 2Pharmaceutical Organic Chemistry Department, Faculty of Pharmacy, Cairo University, Cairo 11562, Egypt; hala.el-nassan@pharma.cu.edu.eg; 3Department of Pharmaceutics, Faculty of Pharmacy and Drug Technology, The Egyptian Chinese University, Cairo 11786, Egypt

**Keywords:** curcumin, wound, healing, bilosomes, hydrogel, chitosan, oxidized alginate

## Abstract

The current study aimed to fabricate curcumin-loaded bilosomal hydrogel for topical wound healing purposes, hence alleviating the poor aqueous solubility and low oral bioavailability of curcumin. Bilosomes were fabricated via the thin film hydration technique using cholesterol, Span^®^ 60, and two different types of bile salts (sodium deoxycholate or sodium cholate). Bilosomes were verified for their particle size (PS), polydispersity index (PDI), zeta potential (ZP), entrapment efficiency (EE%), and in vitro drug release besides their morphological features. The optimum formulation was composed of cholesterol/Span^®^ 60 (molar ratio 1:10 *w*/*w*) and 5 mg of sodium deoxycholate. This optimum formulation was composed of a PS of 246.25 ± 11.85 nm, PDI of 0.339 ± 0.030, ZP of −36.75 ± 0.14 mv, EE% of 93.32% ± 0.40, and the highest percent of drug released over three days (96.23% ± 0.02). The optimum bilosomal formulation was loaded into alginate dialdehyde/chitosan hydrogel cross-linked with calcium chloride. The loaded hydrogel was tested for its water uptake capacity, in vitro drug release, and in vivo studies on male Albino rats. The results showed that the loaded hydrogel possessed a high-water uptake percent at the four-week time point (729.50% ± 43.13) before it started to disintegrate gradually; in addition, it showed sustained drug release for five days (≈100%). In vivo animal testing and histopathological studies supported the superiority of the curcumin-loaded bilosomal hydrogel in wound healing compared to the curcumin dispersion and plain hydrogel, where there was a complete wound closure attained after the three-week period with a proper healing mechanism. Finally, it was concluded that curcumin-loaded bilosomal hydrogel offered a robust, efficient, and user-friendly dosage form for wound healing.

## 1. Introduction

Wounds are breaks and defects in the skin that may occur due to chemical, thermal, or mechanical reasons, leading to a loss of normal skin structure and, consequently, function [1]. Skin injuries are classified as acute or chronic. Acute injury is a skin wound that occurs suddenly and, depending on its depth and size, can be cured within 2–3 months. Acute wounds can heal merely by cleaning and covering the wound, thus relying on the self-healing mechanism of the healthy human body. On the other hand, chronic wounds like burns, infections, or leg ulcers are life-threatening because they do not automatically heal quickly and they require an intervention to accelerate the healing process [2,3].

Wound dressings can significantly speed up the process of wound closure through different collaborating characteristic mechanisms such as absorbing excess wound exudates, protecting the wound from infections and microorganisms, decreasing surface necrosis of the wound, and offering mechanical protection and pain relief [4]. Wound dressings can be classified according to the method of application into primary (directly covering the wound) and secondary (covering the primary dressing) or classified based on the bioactivity (traditional and bioactive) [5,6]. Bioactive wound dressings necessitate the use of a biomaterial (e.g., chitosan (CS) or alginate) [1].

Hydrogels are considered one of the most important and widely used drug delivery systems for wound healing as they have a strong similarity with the native extracellular matrix (ECM) and can provide a humid environment required for wound healing [3,7]. Hydrogels are widely used as wound dressings owing to their wide and feasible characteristics such as their scaffold structure, ability to absorb wound exudates, and ability to transmit moisture and oxygen. Hydrogels may be classified according to the origin of the gelling agent into natural or synthetic [8]. Natural hydrogels (e.g., collagen and gelatin) are biocompatible and cause almost no stimulation of the inflammatory or immunological responses of the body. Synthetic hydrogels (e.g., hydrogels of polyvinyl pyrrolidone or methacrylates) can rehydrate dry skin, aid in autolytic wound debridement, and also may be used for controlled drug release [9,10]. Smart hydrogels were recently developed to overcome some recognized drawbacks of natural hydrogels such as their poor mechanical strength and the probable toxicity of byproducts that may be released after crosslinking reactions. One study suggested the development of smart self-healing chitosan-based hydrogels using oxidized chitosan and hyaluronic acid as safe crosslinkers [11]. Another study also highlighted the use of a smart hydrogel dressing for chronic diabetic wound treatment, where the synergistic effect of catechol groups and epigallocatechin-3-gallate (green tea derivative) reinforced the hydrogel’s characteristic features. This new hydrogel offers adequate mechanical properties, a self-healing capability, and a feasible tissue adhesiveness [12].

Alginate is an anionic polysaccharide that can form hydrogels under very mild conditions, at room temperature, and without the use of toxic solvents [13]. Owing to its gelling ability and high viscosity, alginates and their derivatives are widely used in many pharmaceutical industries. Moreover, chemical modifications of alginate (e.g., oxidation) offer more reactive functional groups on the polymer backbone and consequently attain more versatile pharmaceutical applications as controlled drug delivery systems [14,15]. Alginate hydrogel is one of the most important biomaterials used for wound management because it is capable of absorbing exudates from wounds, a chief requirement for an ideal dressing. Moreover, alginate hydrogels provide painless removal from the wound application site. Also, they are non-toxic, non-allergic, hemostatic, and biocompatible [16]. Additionally, the porosity of the alginate hydrogels allows the entrapment/immobilization of therapeutic agents which are then released at the wound site at a controlled rate [17].

ADA or alginate dialdehyde is the oxidized alginate resulting from the oxidation reaction with the oxidizing agent, sodium periodate on the hydroxyl groups at C-2 and C-3 positions of the uronic units. This structural modification leads to the improvement in the reactive properties and the biodegradation of alginate [18].

Chitosan (CS) is a well-known natural polysaccharide which is a biopolymer of glucosamine and *N*-acetyl-glucosamine units linked by 1–4 glucoside bonds. CS is considered to be the most available, utilized, and distributed biomaterial, second after cellulose [19]. CS is a non-toxic polycationic polymer with a powerful antibacterial property, which can influence all stages of wound healing [20]. CS is known to amplify the function of inflammatory cells; consequently, it promotes tissue organization and granulation; thus, CS is beneficial in the treatment of large and open wounds [21]. CS is considered to be a convenient biomaterial forming drug delivery systems in many pharmaceutical formulations owing to its beneficial properties such as its gelation ability, good tissue adhesive properties, biodegradability, biocompatibility, non-toxicity, and antibacterial effects [22,23]. Alginate dialdehyde (ADA) and/or CS Schiff’s base cross-linkage has been widely used before with or without the aid of cross-linkers. Both ADA [24] and CS [25] are primarily used for the sustained drug delivery of drugs.

Curcumin turmeric is the main active constituent of the plant *Curcuma longa* L. Curcumin has well-recognized antibacterial, antifungal, antioxidant, and anti-inflammatory properties. Moreover, it can induce the endogenous production of transforming growth factor beta 1 (TGF-β1) at the wound application site [20], making it an ideal candidate for wound healing applications. Unfortunately, its low oral bioavailability, poor water solubility, and rapid degradation and metabolism limit its medical applications. In order to maximize the benefits of this miracle molecule, new strategies and technologies have been applied, such as designing curcumin nano-formulations that offer a better way for curcumin to be used for wound healing purposes [26,27].

Drug-loaded nano-systems or nanocarriers (NCs) have been applied widely in the medical and pharmaceutical industries [28]. The use of natural components in the design of NCs is still of high value. Because of the very small particle size (PS) (less than 1 µm), NCs can greatly improve the bioavailability and the sustained-targeted release of the loaded molecules better than microparticles [29]. Therefore, NCs are considered the cornerstone for targeting the whole cycle of wound healing, with their unique properties such as their high surface-area-to-volume ratio, which offers better chances for tissue perfusion and penetration [24,25]. In addition, merging nanotechnology with hydrogel technology will significantly help to boost the wound healing process, as it comprises the summation of both benefits and amplifies their positive effects [1].

Bilosomes are widely used to design delivery systems for different drug candidates and several studies have proved their success and versatile applications [30,31]. The small PS and the high surface area of bilosomes increase the tissue contact time and skin penetration of the active molecules. Moreover, they exhibit the characteristics of optimum nanosystems such as high stability, site-targeted drug delivery, and sustained drug release with reduced side-effects [32].

Many studies have been previously carried out including either curcumin in bilosomes or curcumin in different hydrogel forms for wound healing applications, but the collective benefits of the proposed dosage form give it superiority. Wagleweska et al. tried to use self-assembling bilosomes to deliver two compounds with variable solubility (curcumin and Methylene Blue), but unfortunately, this study lacks in vitro release and in vivo animal studies which are crucial to visualize the release behavior and healing ability of the novel formula [33]. Li et al. studied the development of an in situ injectable nano-composite hydrogel composed of curcumin, N,O-carboxymethyl chitosan, and oxidized alginate as a wound dressing for wound repair application. The results showed that the combined use of nano-curcumin together with N,O-carboxymethyl chitosan and oxidized alginate hydrogel could significantly accelerate the process of wound healing [20]. Marrell et al. investigated the potential of poly(caprolactone) nanofibers as a delivery vehicle for curcumin to be used for wound healing applications. Curcumin-loaded nanofibers enhanced the rate of wound closure despite sustaining the release for only 3 days [34]. In our study, we combined both the benefits of curcumin bilosomes and chitosan-ADA hydrogel to gain all the advantages of this synergistic dressing system in wound healing applications. Cross-linked alginate dialdehyde/chitosan hydrogel loaded with curcumin bilosomes was prepared and it was easy to spread topically and did not require painful interventions as injections. We prepared a nano-liposomal curcumin hydrogel with a sustained curcumin release behavior for 5 days, avoiding the inconvenience of repeated application. The use of ADA/chitosan in the hydrogel reinforces the absorptivity of the formula to wound fluid and exudates.

This work aimed to design and optimize the curcumin-loaded bilosomal hydrogel formulation with specific properties, such as small PS, high surface charge, improved drug encapsulation, and sustained drug release. This would represent a novel wound treatment with bioactive, biodegradable, and biocompatible properties. Curcumin-loaded bilosomal nanovesicles were fabricated using a thin film hydration technique. The optimum formulation was incorporated in a hydrogel using oxidized ADA and CS with the aid of calcium ion cross-linkage. Finally, an in vivo full-thickness excisional wound model was utilized in male Albino rats to investigate the wound-healing activity of the curcumin-bilosomal hydrogel.

## 2. Materials and Methods

### 2.1. Materials

Curcumin was kindly gifted by Guangzhou Phyto Chem Sciences Inc., Guangzhou, China. Sodium alginate (molecular weight, 216.121 g/mol; polydispersity index, 1.22), sodium periodate, ethylene glycol, starch, potassium iodide, disodium hydrogen phosphate, dihydrogen sodium phosphate, and dialysis cellulose membrane (molecular weight cut off, 14,000 g/mol) were all purchased from Sigma-Aldrich, St. Louis, MI, USA. Cholesterol (molecular weight, 384.6 g/mol) and sodium deoxycholate were purchased from Fluka Chemicals, Buchs, Switzerland. Ethanol and methanol were brought from Piochem. Chitosan medium (molecular weight, 190,000–310,000 Da) with a degree of deacetylation (75–85%), Span^®^ 60 (molecular weight, 430.62 g/mol), and sodium cholate were procured from Loba, Mumbai, India. Ascorbic acid and calcium chloride were purchased from El Nasr Pharmaceutical Chemicals, Cairo, Egypt. Ketamine hydrochloride injection, 50 mg/mL, was purchased from Rotex med, Trittau, Germany. Xylazine, xyla-ject (20 mg/mL, 100 mL injectable solution for vet use only, IM injection), was procured from Adwia Pharmaceuticals, New Cairo, Egypt. Sterile biopsy punch needle (size, 12; diameter, 5.0 mm) was obtained from Hospital Equipment Manufacturing Company, Noida, India. The rest of the used chemicals and reagents were of pure analytical grade.

### 2.2. Fabrication of Alginate Dialdehyde (ADA)

#### Oxidation of Sodium Alginate to form Alginate Dialdehyde (ADA)

The oxidation process of sodium alginate (NaAlg) solution was performed at room temperature for 24 h. Briefly, NaAlg (10.00 g) was dissolved in distilled water (600 mL), and then an aqueous solution of sodium periodate (1.769 g in 100 mL distilled water) was added under stirring at 400 rpm for 20 min. Following this, the volume of the mixture was adjusted to 1 L by the addition of distilled water. The reaction mixture was left in a dark place for 24 h at room temperature. Then, the reaction was terminated by the addition of ethylene glycol (3.50 mL) under magnetic stirring (Stuart-320, Delhi, India) at 400 rpm for 0.5 h. The oxidized alginate was obtained by precipitation with the addition of sodium chloride (3.00 g) and ethanol (1 L). Then, the purification step was as follows: the polymer was again solubilized in water (500 mL) and re-precipitated by adding ethanol (500 mL) in the presence of sodium chloride (1.00 g) and dried at room temperature. The formed oxidized alginate was dissolved in distilled water (50.00 mL) and lyophilized (Lyophilizer; Novalyphe-NL 500; Savant Instruments Corp., Holbrook, NY, USA) with the pressure adjusted to 7 × 10^−2^ mbar and the condenser temperature maintained at −45 °C [18,35,36].

### 2.3. Characterization of Alginate Dialdehyde (ADA)

#### 2.3.1. Determination of Oxidation Degree

The oxidation degree of the pre-formed ADA was determined using an indicator method [18]. Firstly, the indicator solution was prepared by mixing equal volumes of 20% *w*/*v* potassium iodide and 1% *w*/*v* soluble starch solutions in phosphate buffer (pH 7). Before adding the quencher (ethylene oxide) into the oxidation reaction, 1 mL of the oxidation reaction solution (NaAlg and sodium periodate with ethanol and distilled water) was diluted to 250 mL with distilled water. Then, 3 mL of this diluted solution was mixed with 1.5 mL of indicator solution and the volume was completed to 5 mL with distilled water. The absorbance of the triiodine–starch complex was rapidly measured with a spectrophotometer (Shimadzu UV—1601 PC, Kyoto, Japan) at 486 nm. Then, the periodate concentration in the sample was deduced using the molar absorption coefficient previously calculated from the absorbance of the complex versus iodate concentration (0.082 M/L). The difference between the initial and final amount of iodate reflected the amount of glycol moieties that were transformed into aldehyde groups [18]. The oxidation percentage was determined using the following equation:Oxidation degree = (consumed amount of periodate in grams/added amount of periodate in grams) × 100

#### 2.3.2. Fourier Transform Infrared Spectroscopy (FT-IR)

FT-IR spectroscopy was performed to check for the new functional groups after the formation of ADA and to compare it with pure NaAlg. Briefly, a sample of 2 mg of each ADA and NaAlg was scanned individually using FT-IR spectrophotometer (Model 8400, Shimadzu, Kyoto, Japan), where the weighed sample was promptly mixed with potassium bromide, compressed, and scanned in the range of 4000–400 cm^−1^ [37].

### 2.4. Formation and Characterization of ADA/Chitosan Hydrogel

#### 2.4.1. Formation of ADA/Chitosan Hydrogel

CS solution (2.5 mL, weight = 4% *w*/*v*) was prepared using ascorbic acid solution 1% *w*/*v* [38]. ADA solution (2.5 mL, weight = 20% *w*/*v*) was prepared using distilled water and then mixed with the prepared CS solution. The mixture was mixed on a hot magnetic stirrer adjusted at 40–50 °C for 10 min to form the CS-ADA hydrogel. Following that, CaCl_2_ (0.10 g) was dissolved in distilled water (10 mL), and then 2 mL of this solution was added carefully to the CS/ADA mixture, forming the cross-linked hydrogel.

#### 2.4.2. Determination of Gelation Time

The gelation time test was performed according to a previously reported method with minor modifications [39]. A mixture of CS and ADA solution treated with CaCl_2_ solution was transferred to a Petri dish (100 × 20 mm^2^), and a magnetic stirring bar (Teflon fluorocarbon resin, 5 × 2 mm^2^) was placed in the center and the solution was stirred at 300 rpm using a hot plate/stirrer adjusted at 40 °C. The gelation time was recorded when the solution formed a solid globule that interfered with the magnetic stirrer bar and separated completely from the bottom of the dish [15].

### 2.5. Formation of Curcumin Bilosomes

Curcumin-loaded bilosomes were formed using the thin film hydration technique [40], with some modifications. Eight formulations were prepared using a simple 2^3^ factorial design (SPSS^®^ software, version 27, SPSS Inc., IBM Corporation, New York, NY, USA). The independent factors chosen were as follows: (A) type of bile salt (sodium deoxycholate, SDC; sodium cholate, SC), (B) amount of bile salt (5 and 10 mg), and (C) cholesterol/Span^®^ 60 molar ratio (1:5 and 1:10 *w*/*w*). The dependent variables measured were as follows: particle size (PS), zeta potential (ZP), entrapment efficiency (EE%), % drug released after 72 h (Q_72h_), and drug content. Briefly, 10 mg of the drug was mixed with a 200 mg mixture of cholesterol and a non-ionic surfactant (Span^®^ 60), as clarified in Table 1. All the components were dissolved in 10 mL absolute ethanol in a 50 mL round-bottomed flask. The organic solvent was evaporated at 40 °C under vacuum utilizing a rotary evaporator (RE 400, Stuart, India) adjusted at 60 rpm for 30 min to allow the complete elimination of organic solvent. The formed thin dry film on the inner wall of the flask was hydrated with 10 mL distilled water carefully, while maintaining normal-pressure conditions and a rotation speed of 60 rpm, using glass beads for 1 h at 40 °C. Following this, the hydrated film dispersion was sonicated for 2 min using a bath sonicator (Hilab, model: GLF 3203, Dusseldorf, Germany) to avoid the formation of any aggregates. SDC was added to the organic phase during the evaporation step and SC was added to the aqueous phase during the hydration step [41].

### 2.6. Characterization of the Prepared Curcumin Bilosomes

#### 2.6.1. Particle Size (PS), Polydispersity Index (PDI), and Zeta Potential (ZP) Determination

PS and ZP of the curcumin bilosomes formulations were assessed after suitable dilution with distilled water (1:10 *v*/*v*), followed by analysis via ZetaSizer Nano ZS (Malvern Instruments, Malvern, UK). The Z-average value was used to represent the PS. Also, PS distributions were assessed through the determination of PDI [42,43].

#### 2.6.2. Drug Content

To assess the drug content of the curcumin bilosomes, 1 mL of each sample was mixed individually with 9 mL ethanol, sonicated for 5 min using a bath sonicator, and then stirred for another 5 min on a hot plate at 40 °C. Actual drug content was measured spectrophotometrically at a wavelength of 424 nm after suitable dilution, and the concentration was concluded in accordance with a preconstructed calibration curve in PBS ethanol/phosphate-buffered saline of 7.4 (30/70) (R^2^ = 0.9998, *n* = 3).

#### 2.6.3. Drug Entrapment Efficiency (EE%)

The EE% of the curcumin bilosomes was assessed by withdrawing a 1 mL aliquot of the bilosomal formulation which was ultra-centrifuged at 15,000× *g* rpm at 4 °C for 60 min using a cooling centrifuge (Hermle Labortechnik GmbH, Wehingen, Germany). The residue was dissolved in 10 mL ethanol under sonication for 5 min in a bath sonicator and then stirred for 5 min to quantify the amount of entrapped drug spectrophotometrically at 424 nm. The drug EE% was determined using the following equation [44]:EE% = (Amount of entrapped drug in mg/Actual drug content in mg) × 100

#### 2.6.4. In Vitro Curcumin Release from Bilosomes

The release of curcumin from bilosome vesicles was performed using the dialysis membrane technique [45]. Certain volumes of bilosomes (equivalent to 2 mg curcumin) in addition to 1.5 mL of release medium were transferred to a dialysis cellulose bag (previously soaked in distilled water for 24 h) and firmly closed from both ends. Each bag was transferred to an amber-colored glass light-resistant bottle (wrapped with aluminum foil) containing 100 mL of release medium, ethanolic phosphate-buffered solution (30% *v*/*v*; pH 7.4), to ensure sink conditions. The glass bottles were placed in a thermostatically controlled shaker (Hilab, model GLF 3203, Germany) operating at 100 rpm and maintained at 37 ± 0.2 °C. At definite time intervals (1, 2, 3, 4, 6, 8, 24, 48, 72, 96, and 120 h), aliquots of 3 mL were withdrawn and immediately replaced by equivalent volumes of fresh release medium. The withdrawn samples were measured spectrophotometrically at 424 nm.

The investigated formulations were compared regarding the percentage of drug released after 2 and 72 h (Q_2h_ and Q_72h_, respectively).

### 2.7. Fabrication of Curcumin Bilosomal Hydrogel

Curcumin bilosomal hydrogel was formed by dissolving 0.4 g of ADA in 2 mL of the selected bilosomal dispersion using a hot plate under controlled temperature (40 °C), adjusted at 300 rpm for 10 min, and then pre-defined volumes of 4% CS and 1% CaCl_2_ solutions were added to the bilosomal dispersion under stirring at 300 rpm using the hot plate for 10 min until hydrogel formation.

### 2.8. Characterization of the Prepared Curcumin Bilosomal Hydrogel

#### 2.8.1. Particle Size (PS), Polydispersity Index (PDI), and Zeta Potential (ZP) Determination

PS and ZP of the curcumin bilosomal hydrogel were assessed using the same method mentioned in Section 2.6.1.

#### 2.8.2. Drug Content and Entrapment Efficiency

The drug content and entrapment efficiency of the bilosomal hydrogel were calculated as mentioned previously in Section 2.6.2 and Section 2.6.3, respectively.

#### 2.8.3. In Vitro Curcumin Release from the Bilosomal Hydrogel

The release of curcumin from bilosomal hydrogel was performed using the dialysis bag technique [46]. The release of curcumin was assessed using the same method mentioned in Section 2.6.4 but with slight modifications, where a certain weight of bilosomal hydrogel (equivalent to 2 mg curcumin) was placed inside the dialysis bag with 100 mL of the release medium.

#### 2.8.4. Water Uptake Capacity of the Hydrogel

The percent of water uptake of the hydrogel was easily determined by a simple method [1]. The selected curcumin bilosomal hydrogel was dried in a hot oven (Titanox, model 63915, Titanox S.r.l, Torre de Picenardi, Italy) at a controlled temperature (40 °C) for 24 h [47], and then the initial dried weight in mg was recorded as (W_o_). After this, the dried hydrogel sample was placed on a sponge pre-soaked in phosphate-buffered saline (pH 7.4) in a container at room temperature. The sample was re-weighed after removing the excess moisture and recorded as (W_t_) at different intervals (4, 8,12, 16, 21, 25, and 29 days). The percent of water uptake (W_s_) was calculated as follows [1,48]:W_s_ (%) = (W_t_ − W_o_/W_o_) × 100

#### 2.8.5. Rheology

The rheological characteristics of the selected curcumin bilosomal hydrogel were assessed using a cone and plate viscometer (Brookfield viscometer; type DVT-2). The plate was connected to a water bath adjusted at 35 ± 0.1 °C. A sample of 0.5 mL of the hydrogel was transferred to the plate. The shear rate was elevated gradually from 0.5 to 100 min^−1^ and the viscosity was recorded accordingly [43].

#### 2.8.6. Differential Scanning Calorimetry (DSC)

Frozen samples of plain and bilosomal hydrogel (−20 °C) were lyophilized for 24 h with the pressure adjusted to 7 × 10^−2^ mbar and the condenser temperature maintained at −45 °C [43].

DSC analysis (DSC 60, Shimadzu, Kyoto, Japan) was performed for curcumin powder, selected bilosomal vesicles, and their physical mixture as well as their individual components. Additionally, the lyophilized plain and bilosomal hydrogel, as well as their individual components, were assessed. In brief, approximately 4 mg of each sample was scanned separately in aluminum pans from 30 to 300 °C with a heating rate of 10 °C/min and an inert nitrogen flow of 25 mL/min.

#### 2.8.7. Fourier Transform Infrared Spectroscopy (FT-IR)

FT-IR spectroscopy was conducted to check for any possible interactions. Briefly, 2 mg of the selected bilosomal formulation, its physical mixture, and the individual ingredients as well as the lyophilized plain and bilosomal hydrogel were scanned using FT-IR spectrophotometer following the same steps as previously mentioned in Section 2.3.2.

#### 2.8.8. Transmission Electron Microscopy (TEM)

Morphological examination of curcumin bilosomes and bilosomal hydrogel was conducted using TEM (Lecia Image, Wetzlar, Germany) connected to camera model TKC 1380 JVC (Victor Company, Tokyo, Japan) adjusted at an acceleration voltage of 80 kV. A sample drop was placed on a carbon-coated copper grid surface and left to dry for 1 min (for F5 dispersion) and 5 min (for the hydrogel); any excess was dried by a tip of filter paper.

### 2.9. Statistical Analysis

All the data were calculated as mean ± standard deviation (*n* = 3). All the results obtained were statistically compared using SPSS^®^ software, version 27 (SPSS Inc., IBM Corporation, New York, NY, USA), using one-way ANOVA followed by least significant difference (LSD) as a post hoc test. The level of significance was *p* ˂ 0.05 in all tested experiments.

### 2.10. In Vivo Animal Study

#### 2.10.1. Animals

The study protocol was reviewed and approved by the Research Ethics Committee of the Faculty of Pharmacy, Cairo University, Egypt. All the used and applied experimental procedures abided by the rules set by the care and use of laboratory animals of the National Institute of Health, issued under the following number: (NIH publication no 85-23, 1996). Eight male Albino rats approximately 8 weeks old and with a weight range of 180–200 g were used in this study. The rats were kept in individual boxes with free food and water access.

The boxes were kept at approximately 25 °C in a well-ventilated room, where it was lit for 12 h and kept dark for another 12 h to match with the natural day and night cycles. The clinical observation of the animals revealed that they were healthy without abnormalities. The animals were categorized into 4 groups. The rats in the first group (A) were treated with curcumin bilosomal hydrogel, while those in the second group (B) were treated with curcumin dispersion (equivalent to 10 mg curcumin). Rats in the third group (C) were treated with plain hydrogel (lacking the addition of drug-loaded bilosomes) and those in the fourth group (D) served as a negative control group that did not receive any treatment.

#### 2.10.2. Surgical Procedure

The wound healing effect of the curcumin bilosomal hydrogel was examined on 8 male Albino rats using a full-thickness wound model and compared to the normal wound healing process (untreated control group). The rats were anesthetized with appropriate doses of ketamine (75 mg/kg) and xylazine (10 mg/kg). Skin shaving, washing with water and soap, and then disinfection with 70% *v*/*v* ethanol took place. Following that, four full-thickness rounded skin excision wounds 5 mm in diameter were performed, two on each side of the spine of every single rat, using a sterile biopsy punch needle (size, 12; diameter, 5.0 mm; Hospital Equipment Manufacturing Company, Noida, India).

The tested samples were prepared under laminar flow. The wounds were covered and taped with elastic gauze during the treatment period. The skin healing process was determined by observing and imaging the rate of wound closure over a three-week interval. All those procedures were carried out under aseptic conditions.

#### 2.10.3. Histopathological Examinations

On the 21st day of the experiment, the rats were euthanized under sodium phenobarbital anesthesia (150 mg/kg intraperitoneal), and all the required efforts were taken to minimize suffering. Full-thickness skin samples from the wound area along with 0.2 mm of peripheral healthy surrounded skin were collected using a dermal biopsy punch needle (size, 12; diameter, 5.0 mm; Hospital Equipment Manufacturing Company, Noida, India). The samples were taken from the skin of rats allocated in different groups and fixed in 10% formaldehyde for 24 h. Washing was performed using tap water and then dehydration was carried out using serial dilutions of alcohol (methyl, ethyl, and absolute ethyl). Specimens were cleared in xylene and embedded in paraffin beeswax at 56 °C in a hot air oven for another 24 h. Paraffin beeswax tissue blocks were prepared for sectioning at 4 µm thickness using a rotary LEITZ microtome (1512; Leitz, Leitz Industries, Wetzlar, Germany). The obtained tissue sections were collected on glass slides, de-paraffinized, and stained with hematoxylin & eosin for examination using a light electric microscope (Leica Microsystems, Wezlar, Germany) [1,49].

## 3. Results and Discussion

### 3.1. Fabrication and Characterization of Alginate Dialdehyde (ADA)

#### 3.1.1. Synthesis of ADA and Determination of the Oxidation Degree

Periodate oxidation using sodium periodate is a common reaction to functionalize the alginate polysaccharide [35]. The oxidation reaction of alginate with sodium periodate is performed on hydroxyl groups at C-2 and C-3 positions of the uronates, generating dialdehyde functional groups on the alginate backbone, as illustrated in Figure 1. Owing to their reactivity, these newly introduced aldehyde groups can interact with the hydroxyl groups present either on the adjacent uronic acid subunits in the same or adjacent polymer chains to form intramolecular and intermolecular hemiacetals, respectively, leading to a scarcity in the available aldehyde groups [18]. The aldehyde groups in oxidized NaAlg can be cross-linked with divalent or even trivalent cations to form strong physical cross-linked hydrogels. Moreover, the free aldehyde on the alginate dialdehyde can react with amine groups—via Schiff’s base reaction—to form chemically cross-linked hydrogels, as illustrated in Figure 1. These cross-linked hydrogels, whether physically or chemically, have versatile biomedical applications such as the delivery of both small drugs and biomolecules to different organs and tissues, to improve the wound healing power [50], and muscle/bone tissue engineering [51,52].

The oxidation degree of alginates is defined as the percentage of oxidized uronic acid groups in the alginate. The oxidation degree in the present work was found to be 77.6%. Still, the newly formed aldehyde group introduced on the alginate polymer offered new reactive groups that were ready for further modification [18,53].

#### 3.1.2. FT-IR

The chemical cross-link mechanism of gelation is attributed to Schiff’s base reaction between the amino group in CS and the aldehyde groups of ADA, as shown in Figure 1.

The FTIR spectrum of NaAlg showed a broad stretching band in the range of 3541–3290 cm^−1^ which was attributed to the hydroxyl group of the residues of mannuronic and guluronic acid. Two other bands could be detected at 2997–2924 cm^−1^ owing to the asymmetric and symmetric CH_2_ stretching. A band at 1631 cm^−1^ was also detected and was related to C=O stretching of the carboxyl group [1,54]. The spectrum of ADA indicated the presence of an OH stretching band at 3414 cm^−1^ and a broad C=O band at 1650–1639 cm^−1^. A characteristic band at 1639 cm^−1^ was still present in ADA, which was attributed to the asymmetric and symmetric vibrational stretching of −COO− groups on the alginate backbone. Therefore, it was concluded that the performed oxidation reaction did not affect the carboxyl groups in alginate. Moreover, the intensities of the symmetrical C–O–C stretching band at 1096 cm^−1^ and 817 cm^−1^ were reduced, confirming the chains’ cleavage. The band that indicated the presence of an aldehyde group was not detected, due to the hemiacetal formation of the free aldehyde groups with the hydroxyl groups on the neighboring uronic acid subunits. These data are consistent with the previously reported FTIR data of oxidized alginate [35,55].

The FTIR spectrum of CS showed a band at 3406–3124 cm^−1^ owing to the overlapping stretching of OH and NH groups. The CH group stretching band was seen at 2920–2877 cm^−1^, while the bands of the amide group C=O and NH bending were detected at 1654 cm^−1^ and 1543 cm^−1^, respectively, as presented in Figure 2.

### 3.2. Formation and Characterization of ADA/Chitosan Hydrogel

Sodium alginate is characterized by massive swelling in water, an unstable morphology, and insufficient mechanical properties which consequently hinder the hydrogel development. These disadvantages were overcome by the use of ADA. In addition, the disadvantages of traditional aldehyde cross-linking agents such as high toxicity and ease of precipitation were also avoided [56]. High-molecular-weight alginate hardly degrades in the human body because of the lack of alginate-degrading enzymes, while alginate with a molecular weight lower than 50 kDa can be removed from the body through kidney excretion mechanisms. The main advantage of alginate oxidation is that oxidation can strengthen its biodegradability to be used as a degradable hydrogel scaffold for the drug delivery system and tissue engineering, where its functional aldehyde groups can be quickly degraded in the body compared with natural alginate [37]. In addition, the oxidized alginate can easily form a Schiff base with many amino-containing polymers like chitosan and gelatin, resulting in the formation of strong hydrogel under mild conditions [57].

CS can be immediately transformed into hydrogel when mixed or homogenized with ADA, which is a multivalent anionic polymer, due to the formation of an intermolecular cross-linkage mediated by electrostatic attraction between CS positively charged amino groups along with the negatively charged carboxylates of ADA, and also due to Schiff’s base formation between the aldehyde groups of ADA and the amino groups of CS (Figure 1).

In the present work, several attempts were carried out to reach the optimum hydrogel composition. Trials revealed that using concentrations of ADA and CS lower than 20% *w*/*v* and 4% *w*/*v*, respectively, could not produce a hydrogel with acceptable appearance and consistency. Also, it was found that the type of acid used for solubilizing CS and preparing its solution had a significant effect on the preparation of the hydrogel. The use of ascorbic acid instead of the widely used acid—acetic acid—was of added value regarding hydrogel characteristics. Ascorbic acid can facilitate the donation of electrons from CS; moreover, it is widely used as an antioxidant and it exhibits anti-inflammatory activity, which can promote immune defense and help to cure wounds [58,59]. In addition, the use of a cross-linker (CaCl_2_ solution) obviously assisted in the formation of hydrogel with a firm 3D network matrix as the calcium ions reinforced the hydrogel via ionic cross-linkage. This resulted in the fabrication of an ionic gelation network with a robust “egg box” structure [60]. This dual cross-linked structure using both covalent and ionic reactions raised from CS/ADA Schiff’s base cross-linkage and calcium ionic gelation, respectively, resulted in a much stronger, well-structured, and rigid hydrogel network. Not only did the calcium chloride reinforce the gelation of the hydrogel, but it also offered many advantages for use in wound healing applications as the formed cross-linked calcium CS/ADA gel was considered a high-absorbent gel. In addition, it has hemostatic properties and was removed smoothly from the wound cavity. Wounds treated with it were characterized by a lower bacterial count [61,62]. The cross-linked polysaccharides showed better mechanical properties and offered platforms with a high flexibility and strength to enhance, adjust, and optimize drug loading and release [63].

#### Gelation Time

The recorded gelation time for the hydrogel prepared using the selected concentrations of ADA (20%) and CS (4%) was found to be 10.50 ± 0.71 min. By examining the formed hydrogel, it was found that it was not firm enough. The addition of CaCl_2_ greatly reinforced the texture of the formed hydrogel; in addition, it significantly reduced the gelation time to 6.75 ± 0.35 min (*p* < 0.05) compared to the one lacking the chemical cross-linker. This reduction in gelation time might be attributed firstly to the formation of strong cross-linkage between CS-ADA as a result of Schiff’s base reaction, and secondly due to the addition of calcium ions resulting in ionic gelation and faster formation of the hydrogel network [1].

### 3.3. Fabrication and Characterization of Curcumin Bilosomes

Curcumin-loaded bilosomes were formed using the thin film hydration technique, with some modifications, where eight formulations were prepared, and the particle size (PS), polydispersity index (PDI), zeta potential (ZP), entrapment efficiency (EE%), % drug released after 2 h and 72 h (Q_2h_ and Q_72h_, respectively), and drug content were determined.

#### 3.3.1. Determination of Particle Size (PS), Polydispersity Index (PDI), and Zeta Potential (ZP)

The mean PS of the bilosomal formulations ranged from 197.10 ± 24.60 (F1) to 680.45 ± 9.65 nm (F8), as presented in Table 1. The obtained nano-sized particles might be attributed to the use of anionic bile salts (SDC or SC), where the steric repulsion between the charged moieties of the negatively charged bile salts arranged on the surface of the bilosomal vesicles caused an increase in the curvature of the vesicles’ membrane, resulting in the formation of nanosized particles [64]. Statistical analysis of the investigated independent variables (A, B, and C) showed their significant effect on PS (*p* < 0.01). Regarding the type of bile salt used (Factor A), it was evident that the use of SC (F2, F4, F6, and F8) resulted in a significantly larger PS compared to SDC (F1, F3, F5, and F7), as presented in Table 1 and illustrated in Appendix A. This may be attributed to the difference in the structure of the two bile salts used, where the bulkier SC (430.6 g/mol) led to the formation of bilosomes with a larger PS in comparison to the less bulky SDC (414.56 g/mol) [65]. This difference between SDC and SC formulations might also be ascribed to the presence of one additional OH group in SC compared to SDC, which exerted an additional steric hindrance on the formed vesicle surface and resulted in the difference in the PS values of the two types of vesicles [66,67].

Concerning the amount of the bile salt used (Factor B), it was found that bilosomes prepared using the lower amount of the bile salt (5 mg) showed a significantly smaller PS (*p* ˂ 0.01) compared to the higher amount (10 mg) as plotted in the Appendix A. This reduction in PS as a function of the lower bile salt content in the lipid bilayers might be explained according to the Helfrich interactions present between bilayers in the membrane stack, where the subsequent increase in PS upon the use of the higher bile salt content may be due to the bilayer saturation. The bilosome bilayers are flexible and do not exhibit rigid planes, so they experience thermally excited bending fluctuations that generate steric repulsions between the colliding bilayers. This repulsive interaction makes it possible to incorporate the increasing amount of bile salt so that the inter-bilayer equilibrium distance increases and, consequently, a larger amount of water would enter the core of the vesicles during the dispersion step, leading to a size increase in the formed vesicles [68].

By investigating the effect of the cholesterol/Span^®^ 60 ratio (Factor C), it was found that increasing the surfactant proportion in the cholesterol/Span^®^ 60 mixture (moving from cholesterol/Span^®^ 60 ratio of 1:5 to 1:10) had significantly increased the PS (*p* < 0.01), as shown in Table 1 and Appendix A. This might be attributed to the surfactant structure, where the diameter of the nanovesicles was closely dependent on the length of the alkyl chain of the surfactant used. Therefore, surfactants with longer alkyl chains like Span^®^ 60 generally produced large-sized vesicles and so increasing its concentration would consequently increase the PS of the formed bilosomes [69].

PDI assures the uniformity of the PS distribution, where PDI > 0.7 indicates a high aggregation and heterogenous size distribution of the formulated vesicles and a consequently lower stability of the dispersion [70]. In this work, the obtained PDI values ranged from 0.339 ± 0.030 to 0.650 ± 0.070, which proved the homogeneity of the formulations. The bilosomal formulations dispersions had a smooth and flexible texture and showed no blooming.

It has long been negotiated that carrier surface charge influences the vesicular transport across several biological barriers [71]. The ZP values of the prepared curcumin bilosomes were between −28.60 ± 0.56 and −38.90 ± 0.28 mV. These findings confirmed that the formed bilosomes had a sufficient negative charge to maintain the electric repulsion between them and avoid their aggregation, and hence a better stability. It has been proven that the particles with a ZP of ±30 mV or higher are considered stable [72,73]. The negative value of the surface charge on all formulations was mostly due to the use of the negatively charged bile salts (SC and SDC). Statistical analysis on three independent factors (A, B, C) showed that factors A and B had a significant effect on ZP (*p* < 0.05), while factor C showed a non-significant effect on ZP (*p* = 0.43).

It was revealed that the formulations prepared using SDC had significantly higher absolute ZP values than those prepared by SC, as shown in the Appendix A. This difference in ZP values between SDC and SC formulations might be attributed to the presence of one additional OH group in SC compared to SDC, which might lead to the difference in the ZP values [66,67,74,75]. By examining the effect of the amount of bile salt used (Factor B), it was also obvious that increasing the amount of bile salt from 5 to 10 mg significantly increased the ZP (*p* < 0.01) value, as shown in Appendix A due to the increase in the anionic charge on the prepared vesicles.

#### 3.3.2. Drug Content

The bilosomal formulations showed drug content values ranging between 85.15 ± 0.21 and 98.20 ± 0.28%, as presented in Table 1. This considerably high drug content proved a decreased drug loss during the preparation of bilosomes, confirming the suitability of the bilosome preparation method and the components used in enclosing curcumin.

Factorial analysis was carried out on the three independent variables (A: type of bile salt, B: amount of bile salt, and C: cholesterol/Span^®^ 60 ratio) and the results showed that there was no significant difference between the tested bilosomal formulations (*p* > 0.05), as shown Appendix A.

#### 3.3.3. Entrapment Efficiency (EE%)

The EE% of the formed bilosomes ranged between 86.60 ± 0.07 and 97.75 ± 0.1%, as presented in Table 1. Factorial analysis showed that factors A, B, and C possessed significant effects on EE% (*p* < 0.01).

By testing the type of bile salt effect on EE% values (Factor A), it was noticed that the calculated curcumin EE% upon using SDC, which ranged from 92.50 ± 0.14 to 97.75 ± 0.10%, was significantly higher (*p* < 0.01) when compared with SC formulations (ranged from 89.50 ± 0.07 to 85.20 ± 0.07%), as presented in Table 1 and Appendix A. This associated significant increase in curcumin EE% when using SDC might be ascribed to its smaller molecular size (414.6 g/mol) and hence less steric hindrance upon closure of the particles [64,76]. In addition, SDC increased the EE% because it can be incorporated into the bilayer membrane surface, and it increased the flexibility of the lipid membrane and, consequently, increased the solubility of the drug in the lipid membrane, leading to an increased drug entrapment efficiency [77,78]. Likewise, SDC had a lower critical micelle concentration and stronger hydrophobic nature than SC which aided in increasing the EE % of hydrophobic drugs like curcumin [79].

The reduced EE% values associated with the use of SC might be attributed to the elevated hydrophilic–lipophilic balance (HLB) of SC compared to SDC [80,81]. The elevated HLB of SC might assist in the formation of micelles in the dispersion medium as well as fluidizing the lipid bilayer membrane, which resulted in a reduction in curcumin EE%. SC-based bilosomes possessed lower EE% values compared to SDC-based ones because SC at certain concentrations disrupted the organized vesicular membrane bilayer structure, resulting in leakage and loss of the drug from the nanovesicles [82]. Also, SC may displace some curcumin from the bilayer due to structural similarity [83].

It was obvious that the higher bile salt amount (10 mg) showed a lower curcumin EE%. This might be attributed to the fluidizing effect of bile salts on vesicle bilayers at high concentrations, and this might have led to the leakage of the entrapped curcumin from bilosome vesicles, as shown in Appendix A [84].

Increasing the ratio of cholesterol/Span^®^ 60 (Factor C) from 1:5 to 1:10—in other words, increasing the surfactant proportion—significantly decreased the curcumin EE% (*p* < 0.01), as shown in Table 1 and Appendix A. This was attributed to the possible existence of mixed micelle systems in the dispersion medium, which consequently increased the drug solubility in the dispersion medium, therefore lowering the EE% [76]. Also, decreasing the amount of cholesterol in the bilosome composition may decrease the rigidity of the vesicular membrane, leading to drug leakage from the vesicular structure [78].

#### 3.3.4. In Vitro Drug Release

The experiment of the in vitro release of curcumin was set in ethanolic/phosphate-buffered saline (pH = 7.4) to simulate the wound environment [45]. As illustrated in Figure 3a, it was evident that >90% of curcumin was released from its free dispersion within 6 h, which ensured that the drug could simply diffuse through the dialysis membrane used. Regarding the release of curcumin from the bilosomal formulations, a burst drug release (Q_2h_) was shown and it ranged from 11.19 ± 0.70% to 38.42 ± 0.14%, followed by a sustained release profile over 72 h. The results of Q_2h_ and Q_72h_ values are shown in Table 1.

As illustrated in Table 1, it was evident that the Q_2h_ values of curcumin from the bilosome formulations were between 11.19 ± 0.70 and 38.42 ± 0.14%, showing a rapid drug release due to the presence of adsorbed curcumin on the bilosomal surface [85,86]. Regarding the Q_72h_ values, the percentage drug release was between 28.13 ± 0.14 and 96.23 ± 0.14%. The sustained release behavior may be due to the high EE% of curcumin in the prepared bilosomal vesicles (ranged from 85.67 ± 0.34% to 96.59 ± 0.10%). Moreover, the lipophilic nature of the bilosomal vesicles can behave as a drug reservoir, resulting in a slow drug release [1,76].

For the optimum use of curcumin in wound healing applications, a burst drug release, offering a considerable concentration of the drug at a rapid rate at the target site, followed by a sustained drug release, promotes gradual healing of the wound and is considered the best practice in treating skin wounds [87].

Statistical analysis of Q_72h_ revealed that factors A, B, and C showed a significant effect (*p* < 0.01) on Q_72h_. Formulations prepared using SDC showed a significantly higher Q_72h_ compared to SC formulations, as shown in Appendix A. Employing the lower-molecular-weight bile salt SDC during bilosome preparation led to the formation of a larger surface-area-to-volume ratio and resulted in a faster drug release from the nanovesicles [69].

Moving to factor (B), increasing the amount of bile salt from 5 mg to 10 mg resulted in a significantly lower Q_72h_, as shown in Appendix A. This was probably due to the increase in the PS range and decrease in total surface area exposed to the release medium. Additionally, the increased negatively charged molecules of bile salts stabilized the bilosomal bilayers, resulting in the complete inclusion of drugs inside those bilayers, thus reducing the release rate [71].

Changing the cholesterol/Span^®^ 60 ratio from 1:5 to 1:10 significantly (*p* ˂ 0.01) increased the amount of curcumin released from the prepared bilosomes at 72 h (Q_72h_), as shown in Appendix A. This was probably due to the higher proportion of Span^®^ 60 in the vesicles’ composition, which increased the drug solubility and thus promoted the amount of curcumin released from the bilosomes [88].

From the previous results, it was concluded that formulation F5 (prepared using 5 mg of SDC and at cholesterol/Span^®^ 60 ratio of 1:10) was the best formulation. This bilosomal formulation showed a PS of 246.25 ± 11.85 nm, PDI of 0.339 ± 0.030, and high ZP (−31.75 ± 0.35 mV) and EE% (93.32 ± 0.40%) values. Although this formulation exhibited a burst drug release after 2 h (Q_2h_ = 38.42 ± 0.14%), it showed the highest curcumin release after three days (Q_72h_ = 96.22 ± 0.70%). Accordingly, formulation F5 was chosen as the optimum bilosomal formulation and was further loaded into the hydrogel dosage form.

### 3.4. Fabrication and Characterization of Curcumin Bilosomal Hydrogel

Images of plain and medicated (curcumin-loaded bilosomes) hydrogels are presented in Figure 3b,c respectively. A diagram illustrating bilosomal hydrogel containing curcumin-loaded bilosomes and cross-linked using calcium ions is shown in Figure 3d. Interestingly, the curcumin-loaded bilosomal hydrogels combined the dual characteristics of both hydrogels and nanovesicles—the sustained drug release, ease of topical application, and high contact time of the hydrogel to the wound tissue—along with the numerous advantages of the bilosome nanosystem such as the increased encapsulated curcumin solubility, stability, and permeation across biological barriers, which consequently supported the drug bioavailability and pharmacological effects [89].

#### 3.4.1. Determination of Particle Size (PS), Polydispersity Index (PDI), and Zeta Potential (ZP)

As shown in Table 1, the mean PS of the selected formulation, F5, loaded in the hydrogel was 212.00 ± 25.03 nm, which is optimum for wound healing applications. Also, the mean PDI and ZP were 0.434 ± 0.050 and ZP −54.00 ± 2.82 mV, respectively.

One-way ANOVA statistical analysis was conducted, and no significant differences could be detected between the PS and PDI of the free bilosomal dispersion, and the incorporated bilosomal hydrogel (*p* = 0.102 and 1.29, respectively). On the other hand, a significant difference was detected in ZP (*p* = 0.008). This increase in ZP may be due to the use of the negatively charged oxidized alginate polymer in the hydrogel formulation [90].

#### 3.4.2. Drug Content

The reassessment of the drug content of the curcumin bilosomal formulation F5 after its incorporation in the hydrogel showed some drug loss when compared to the free bilosomal dispersion. The drug content in the bilosomes before loading into hydrogel was 95.16 ± 0.03%, while after the incorporation of bilosomes in hydrogel, the drug content decreased to 91.72 ± 0.16%, though this difference was not significant (*p* > 0.05).

#### 3.4.3. Entrapment Efficiency (EE%)

The mean EE% of curcumin in bilosomal hydrogel and free bilosomes (F5) was 90.55 ± 0.21% and 93.32 ± 0.40%, respectively, as presented in Table 1. ANOVA analysis showed a significant decrease in EE% (*p* = 0.003) after loading the curcumin-loaded bilosomes in ADA/chitosan hydrogel. This decrease in EE% may be attributed to the slight decrease in the drug content of the bilosomal hydrogel. This may be due to the presence of the cross-linking ionic agents (CS and calcium) in the formed hydrogel. It was previously assumed that during the cross-linking process, associated water loss can take place, which can result in the loss of drug molecules [91,92].

#### 3.4.4. In Vitro Drug Release

As illustrated in Table 1 and Figure 3a, it was shown that curcumin-loaded bilosomal hydrogel showed a sustained release behavior for 5 days where about 100% of the drug was released compared to the free F5 bilosomal formulation which showed an almost complete release after 3 days only. The Q_2h_ and Q_72h_ were 14.55 ± 0.14% and 74.08 ± 0.70% versus 38.42 ± 0.14% and 96.23 ± 0.02%, for the hydrogel and F5 formulation, respectively. Statistical analysis showed a significant difference (*p* < 0.01) upon comparing Q_2h_ and Q_72h_ of the F5 bilosomal formulation and curcumin-loaded bilosomal hydrogel. The incorporation of the free bilosomes in the hydrogel managed to decrease the initial burst release of curcumin. The following slower drug release pattern was due to the highly lipophilic nature of nanovesicles, which performed as drug reservoirs and released the enclosed drug slowly. In addition, the ability of the hydrogel to absorb more water from the surrounding environment, leading to its swelling and formation of the 3D gel network, limited the access of the dissolution medium and thus attained a profound sustained drug release [93,94]. Also, the uniform inter-network matrix of the hydrogel could easily and steadily release the drug out of the hydrogel in a controlled manner.

This type of drug-controlled release is considered advantageous when used in wound healing applications. An ideal wound treatment must have a sustainable degradation rate to control the drug release and subsequently reduce the treatment replacement frequency [1,95]. The enhanced sustained release of the curcumin proved that the bilosomes were successfully included inside the hydrogel cross-linked polymeric chains, where the drug needed to cross through two different barriers to be released at the wound tissue site, which explained the sustained and controlled release pattern as discussed before [1,95].

#### 3.4.5. Water Uptake Capacity

There is no doubt that determining the water absorption capacity of hydrogel is of great importance in wound treatment applications to predict how the wound dressing will act when applied in a moist wound environment. The absorptivity of the tissue exudates of highly vascularized wounds is certainly a significant feature for an efficient wound treatment, which guarantees the supply of oxygen and nutrients to the injured area [96,97].

Figure 4a shows the gradual increase in the percent of water uptake by the hydrogel reaching its maximum (729.50 ± 43.13%) after four weeks. After this period, the hydrogel gradually lost its rigid and firm appearance. Initially, when the hydrogel was placed on the saturated sponge with phosphate-buffered saline, there was a replacement of the calcium ions of the cross-linked hydrogel with the sodium ions in phosphate-buffered saline. This replacement resulted in the relaxation of alginate cross-linked chains, leading to a massive increase in water absorption and swelling of the hydrogel matrix. As time progressed, many more replacements took place which further led to more swelling and disruption of those entangled alginate polymeric chains, and finally, the hydrogel lost its cross-linked integrated shape and completely broke up into fragments [1].

#### 3.4.6. Rheology

After recording and plotting the average viscosity of bilosomal hydrogel with the shear rate as presented in Figure 4b, it was noticed that the initial gel viscosity was high (63,970.00 ± 1.14 cP), and by increasing the applied shear rate, the viscosity decreased, which indicated the shear thinning or pseudoplastic behavior of the bilosomal hydrogel [98]. This initial high viscosity might be due to the presence of ADA in the hydrogel matrix, which accelerated the transformation of the cross-linked CS solutions from a liquid-like state to a gel-like state [99]. It is worth mentioning that the pseudoplasticity gel performance is beneficial for topical formulations that are obviously viscous under static conditions, where it becomes less viscous after mild hand-rubbing (shear stress application), resulting in an improved spreadability and hence improved drug delivery [100,101].

#### 3.4.7. Transmission Electron Microscopy (TEM)

Photomicrographic images of the two examined samples, free bilosomes (F5) and bilosomal hydrogel, showed non-aggregating spherical vesicles with well-defined walls and smooth surfaces, as shown in Figure 4c,d, respectively. This was a good indication that the incorporation of the optimal curcumin-loaded bilosome formulation (F5) in the hydrogel did not affect their shape.

#### 3.4.8. Differential Scanning Calorimetry (DSC)

DSC was performed to determine the exact melting points of the reagents used and to compare them with the reported data to ensure their purity before using them in the formulation. Therefore, only the heating cycle was needed to determine the melting point of each compound.

The DSC thermogram of curcumin showed a sharp endothermic peak at 160.33 °C, which indicated the melting point of its crystalline form [45,102]. The Span^®^ 60 DSC thermogram revealed an endothermic peak at 59.12 °C, representing its transition temperature [103,104]. SDC gave an exothermic peak at 186.85 °C [105]. Cholesterol exhibited an endothermic peak, indicating its melting point at 148 °C [106,107].

The physical mixture retained most of the peaks of the individual constituents, but a decrease in their intensity could have taken place due to the dilution effect during mixing, with no evidence of interaction between curcumin and the excipients used. The bilosome dispersion showed the disappearance of the characteristic peaks of curcumin and other ingredients, and this disappearance confirmed the presence of curcumin in amorphous form and the formation of a new matrix nano-system.

The DSC thermogram of unloaded hydrogel and bilosomal hydrogel showed that the peaks of ADA, curcumin, and all other ingredients were nearly diminished. This complete disappearance of peaks confirmed the formation of a new matrix (nano) system as presented in Figure 5.

#### 3.4.9. Fourier Transform Infrared Spectroscopy (FT-IR)

The IR spectra of curcumin, cholesterol, Span^®^ 60, SDC, curcumin bilosomal formulation F5, the physical mixture, and loaded and unloaded curcumin bilosomal hydrogel after lyophilization are presented in the Appendix A.

The IR spectrum of curcumin showed a noticeable stretching band at 3510 cm^−1^ owing to its characteristic phenolic OH group. Some other stretching vibrations were also obvious at 1627 cm^−1^ and 1597 cm^−1^ due to C=O and C=C, respectively [45,108]. The IR spectrum of cholesterol showed the characteristic absorption peak of the OH group at 3402 cm^−1^. The bands at 2939–2886 cm^−1^ were attributed to the CH_2_ and CH_3_ group’s symmetric and asymmetric stretching vibration. Other bands such as 1465 cm^−1^ and 1377 cm^−1^ were related to the C-H group bending vibrations [109,110].

The IR spectrum of Span^®^ 60 displayed a strong OH stretching band at 3414 cm^−1^ together with the aliphatic CH stretching band at 2916–2860 cm^−1^ and an ester C=O band at 1739 cm^−1^ [110,111]. The SDC showed the characteristic broad band of OH stretching at 3383 cm^−1^. Two other bands were detected at 2939 cm^−1^ and 2866 cm^−1^, respectively, related to the asymmetric and symmetric CH_2_ vibrational stretching [1,112].

The IR spectrum of the physical mixture indicated the presence of all the characteristic bands of the components without any signs of interaction between them; however, a slight reduction in their intensity could have taken place due to the dilution effect that occurred during mixing [45]. The bilosome dispersion gave a similar IR spectrum with no sign of interaction between curcumin and the components forming the bilosomes (cholesterol, Span^®^ 60, and SDC).

The IR spectrum of the plain hydrogel (CS and ADA) indicated the presence of a OH band at 3417 cm^−1^ and C=O band at 1643 cm^−1^_._ A new band appeared at 1639 cm^−1^ attributed to C=N and provided evidence for the formation of Schiff’s base between CS amino groups and ADA aldehyde groups [35]. Similarly, the hydrogel loaded with the curcumin bilosomes revealed the presence of the characteristic bands of its components. The low intensity of the C=O bands here might be attributed to the presence of bilosome components in very small amounts [1]. All the above data indicated that no interaction occurred between the curcumin and the components of the bilosomes or the hydrogel.

### 3.5. In Vivo Animal Study

There was no death of animals distributed among different groups throughout the time of the experiment. This study was based on measuring the percentage of wound size reduction at specific time intervals as well as examining the tissues’ histopathological features. The wound healing power of curcumin-loaded bilosomal hydrogel (A) was investigated and compared to the effect of other tested groups: curcumin dispersion (B), plain hydrogel (C), and untreated control (D).

#### 3.5.1. Rate of Wound Closure

The skin healing process of the different treatments employed in rats of the different groups was determined macroscopically by observing the rate of wound closure along a three-week time interval as shown in Figure 6 and Figure 7. Through the first week, it was noticed that pus, bleeding, and wound exudates were still present in the untreated control group D, while those features were resolved in the other three groups receiving the other treatments (Figure 6). Similarly, the images of the second and third weeks revealed the progressive healing process of the wound in rats of group A and to a lesser extent in rats of groups B and C, while those in group D maintained the same wound features and were completely unhealed.

The reason for the absence of signs of wound contaminations in groups A, B, and C may be ascribed to the antimicrobial properties of CS and/or the presence of curcumin with its broad-spectrum antibacterial action against a wide range of Gram-positive/Gram-negative bacteria [113,114].

Statistical analysis of the wound size reduction after one week post treatment revealed that the skin wounds in groups A, B, and C showed a significantly smaller wound size (0.15 ± 0.002 cm, 0.35 ± 0.007 cm, and 0.3 ± 0.014 cm, respectively) compared to control group D (0.4 ± 0.014 cm) (*p* ˂ 0.05) (Figure 7). In addition, comparing groups, A, B, and C with each other revealed that group A was statistically significant (*p* ˂ 0.01) compared to groups B and C, whereas groups B and C showed non-statistical significance from each other (*p* = 0.09).

It was obvious that the treatment used in rats of group A exhibited the best healing power. This can be related to the important role of curcumin in improving the rate of wound contraction, suppressing the inflammatory response, enhancing collagen deposition, and promoting angiogenesis, which consequently accelerated the healing rate of wounds [115]. Concerning the treatment results for animals in group C, CS was the determining factor in the healing process, as it could enhance the formation of fibroblast, macrophage, and leucocyte cells that were involved in the wound closure process [21,116]. Moreover, the use of ADA in the hydrogel was regarded as support for the wound healing ability, as it enhanced the hydrogel water vapor transmission capacity; thus, they can provide the moist environment needed for treatments used in wound healing applications [95,117].

Figure 6 showed that at the second-week post-treatment time point, skin wound sizes in groups A, B, and C showed a significant decrease (*p* ˂ 0.01) compared to their counterparts in the first week. Moreover, after comparing the results of the treatment of rats in group D in the first week with those after the second week (0.4 ± 0.014 cm and 0.35 ± 0.014 cm, respectively), it revealed a non-significant difference (*p* > 0.07). Wound sizes were 0.01 ± 0.15 cm, 0.25 ± 0.07 cm, and 0.15 ± 0.15 cm for groups A, B, and C, respectively. By comparing the results of the treatment of rats in groups A, B, and C with each other, it was deduced that the treatment of rats in group A showed a significant statistical reduction in wound size (*p* ˂ 0.01) compared to those in groups B and C. Moreover, the treatment of rats in group C possessed an enhanced wound reduction than those in group B (*p* = 0.03). The superiority of the treatment of rats in group C to that used in group B at this time point was due to the biomaterial CS/ADA used in the hydrogel formation, which act as an efficient wound treatment by offering anti-bacterial action together with absorbing wound exudates needed to initiate the healing process of wounds [118].

By the end of the third week of treatment, wounds in groups A, B, and C were completely closed with the disappearance of the wound gap (Figure 6), leaving only a slight line scar. Wounds treated with only the curcumin dispersion (group B) and hydrogel (group C) showed a larger scar than those treated with curcumin bilosomal hydrogel (group A). On the other hand, control group D showed an incomplete wound closure with a wound size of 0.35 cm ± 0.014 and an obvious long-line scar.

Curcumin had antibacterial, anti-inflammatory, and antioxidant actions and it accelerated the proliferation and remodeling stages over the tissue regeneration process. In addition, the polymer composition CS/ADA of the hydrogel boosted the antimicrobial action of the formulation. In addition, the cross-linked polymer of calcium CS/ADA absorbed the wound exudates and served as a scaffold, helping to initiate re-epithelization and granulation of the tissue. All these factors indicated the synergistic role of curcumin and the cross-linked calcium CS/ADA hydrogel as a treatment for wounds, with the enhanced wound size reduction in rats of group A supporting this finding.

#### 3.5.2. Histopathological Examinations

Histopathological examinations were carried out to support the results of wound size reduction and to make sure that the healing of tissues followed the appropriate pattern. The third-week post-treatment microscopical examination of wound tissue sections is shown in Figure 8.

Skin samples of rats in group A (curcumin-loaded bilosomal dispersion) (Figure 8a) showed an epidermis with a thin, newly formed epidermal layer (curvy arrow) with the existence of a noticeable number of migrating epidermal cells (arrow) besides the noticeable progress in skin gland development (rectangle). The dermis layer demonstrated well-organized thick bundles of fibrous connective tissue (arrow with tail), or thin fibers in an irregular distribution (star), and partial areas with interstitial edema (wave arrow) besides the infiltration of a low number of inflammatory cells (arrowhead), which indicated a successful and proper ongoing healing process and complete dermis formation.

Skin samples of rats in group B (curcumin dispersion) (Figure 8b) displayed an epidermis with a newly formed epidermal layer (curvy arrow), few migrating epidermal cells (arrow), as well as a low number of newly developed skin glands (rectangle), a dermis with a well-developed amount of fibrous connective tissue (arrow with tail), the infiltration of a moderate number of inflammatory cells (arrowhead), and interstitial edema leading to dispersion between the fibrous connective tissue (wave arrow).

Skin samples of rats in group C (plain hydrogel) (Figure 8c) showed a marked epidermis also with a freshly formed epidermal layer (curvy arrow); the emergence of moderate numbers of migrating epidermal cells (arrow) plus a moderate number of newly progressed skin glands (rectangle); a dermis with well-developed, thick bundles of collagen fibers (arrow with tail); the infiltration of a moderate number of inflammatory cells (arrowhead); and an obvious amount of interstitial edema, leading to dispersion between the fibrous connective tissue (wave arrow).

Skin samples of rats in group D (untreated control) (Figure 8d) revealed an epidermis layer with a developed attached scab above the wound surface (curvy arrow). Moreover, there was the presence of a dermis with a parallel organization of fine fibrous tissue (star), as well as an intense interstitial infiltration of inflammatory cells (arrowhead).

Those findings proved the efficient role of curcumin-loaded bilosomal hydrogel (Group A) in the wound healing process, compared to the other treatments for rats in groups B and C, which showed incomplete tissue structures, as well as group D, which presented a very large number of inflammatory cells and edema, which indicated an improper wound healing mechanism.

Curcumin is known to have many characteristics that enforce its use in wound healing applications as antibacterial, antifungal, antioxidant, and anti-inflammatory properties [20]. The addition of curcumin bilosomal nanovesicles in the CS/ADA hydrogel accelerated the wound healing process and the construction of normal skin tissue layers. CS has antibacterial properties, which can augment all stages of wound healing [20]. CS is known to amplify the function of inflammatory cells, consequently promoting tissue organization and granulation [21]. The cross-linked calcium CS/ADA hydrogel aided wound management due to its capability of absorbing massive wound exudates as well as cooling the surface of the wound [16,119]. Moreover, the hemostatic action of calcium ions enhanced the wound healing mechanism [120]. Also, the small nano-size of the bilosomal formulation besides its high curcumin EE% and surface area increased its penetration. Finally, the sustained drug release over 5 days from the bilosomal hydrogel may explain the superior action of the medicated hydrogel over the investigated samples.

## 4. Conclusions

This study presented a simple technique for the fabrication of curcumin-loaded bilosomal hydrogel used in wound healing applications. Curcumin bilosomal hydrogels offered many advantages, which made them an efficient treatment for wound healing for various aspects. ADA was fabricated through the oxidation of sodium alginate and was then combined with chitosan to form a hydrogel with a gelation time of 6.750 ± 0.353 min. Curcumin bilosomes were prepared and the best formulation composed of 5 mg of SDC and a cholesterol/Span^®^ 60 ratio of 1:10 possessed a PS of 246.25 ± 11.85 nm, PDI of 0.339 ± 0.030, ZP of −36.75 ± 0.14 mV, drug content of 95.12 ± 0.03, EE% of 93.32 ± 0.40, and Q_72h_ of 96.23 ± 0.02 was incorporated in the prepared hydrogel.

The curcumin-loaded bilosomal hydrogel was able to sustain the curcumin release over five days. The hydrogel showed a high % of water uptake at the four-week time point (729.50 ± 43.13%) as well as shear-thinning behavior. The animal study showed that after three weeks of wound induction and treatment, groups A, B, and C showed completely closed wounds with a slight line scar. Wounds treated with only the curcumin dispersion (group B) and hydrogel (group C) showed a slightly larger scar than those treated with curcumin bilosomal hydrogel (group A). On the contrary, control group D showed incomplete wound healing and closure with a wound size of 0.35 mm ± 0.014 and an obvious long-line scar. Histopathological studies proved the superior behavior of the curcumin-loaded bilosomal hydrogel in wound healing where the wounds were completely closed after three weeks with programmed healing mechanism. This research presented the development of curcumin-loaded bilosomal alginate dialdehyde/chitosan hydrogel as an efficient, affordable, and promising wound treatment. Finally, the success of curcumin-loaded bilosomal hydrogel may be because of the synergistic action of curcumin-loaded bilosomes along with the cross-linked alginate dialdehyde/chitosan hydrogel.

## Figures and Tables

**Figure 1 pharmaceutics-16-00090-f001:**
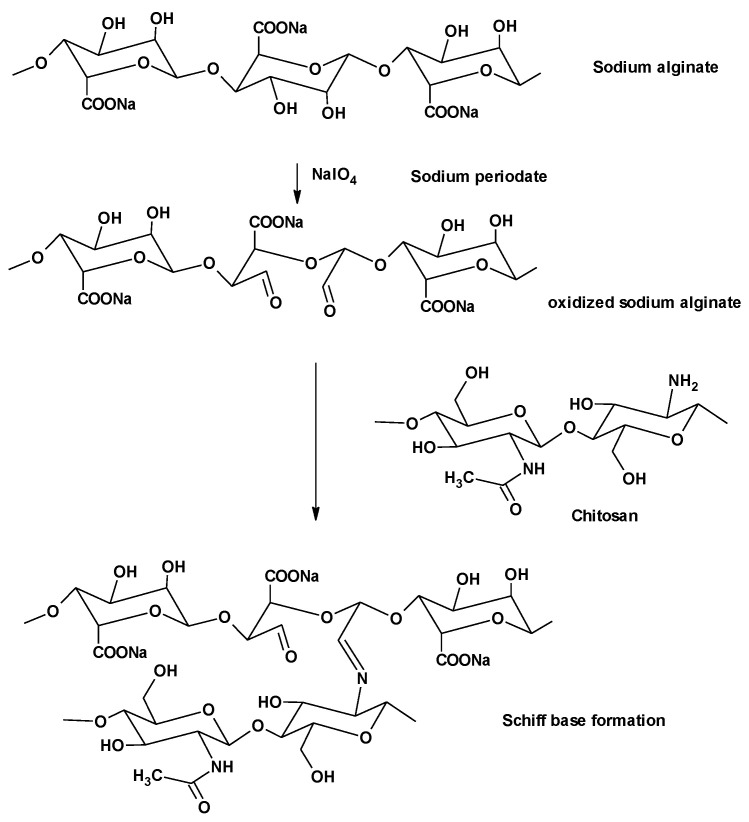
Oxidation of sodium alginate and reaction of alginate dialdehyde with chitosan.

**Figure 2 pharmaceutics-16-00090-f002:**
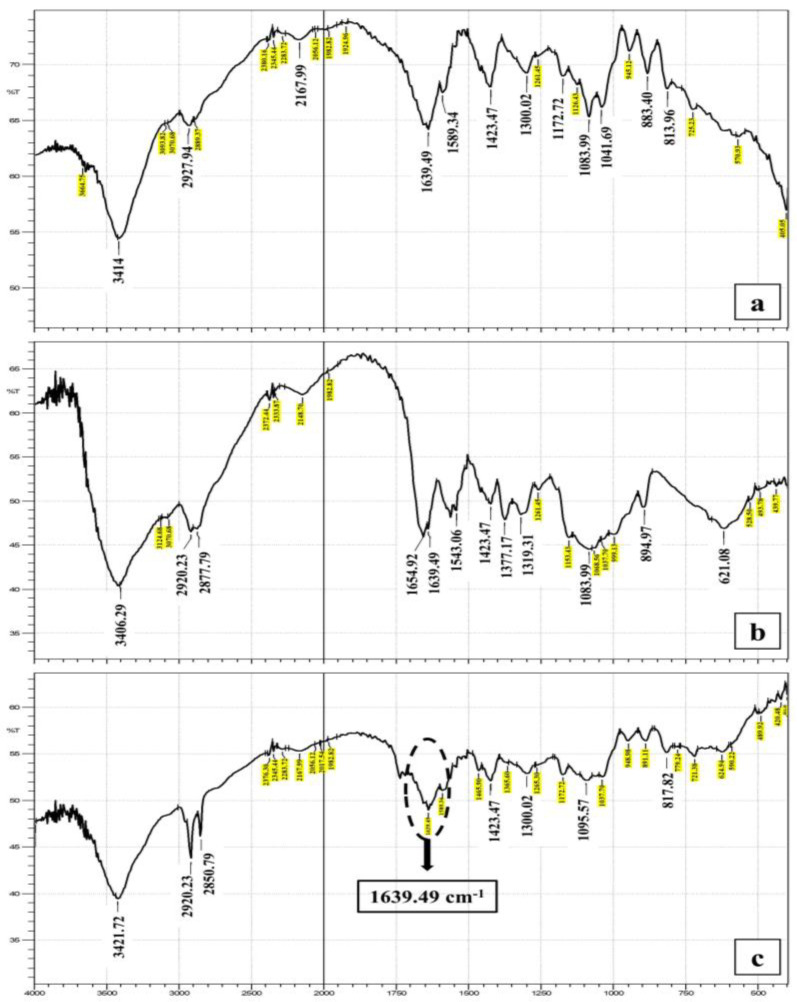
Collective FT-IR graphs of (**a**) alginate dialdehyde, (**b**) chitosan, and (**c**) alginate dialdehyde—chitosan hydrogel shows a peak at 1639 cm^−1^ indicating C=N bond formation in the Schiff’s base.

**Figure 3 pharmaceutics-16-00090-f003:**
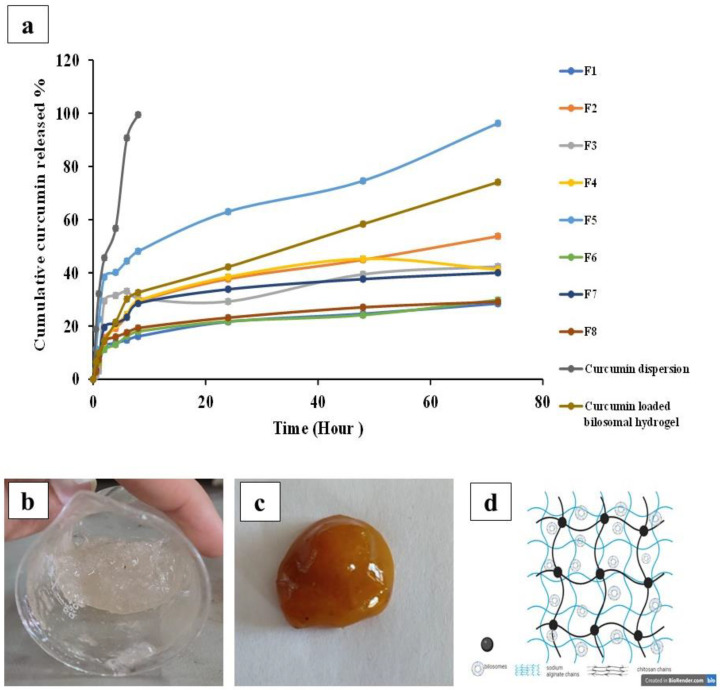
(**a**) Mean curcumin release profiles from different bilosomal formulations (F1 to F8), curcumin dispersion, and curcumin-loaded bilosomal hydrogel through dialysis membrane in ethanolic/phosphate-buffered saline of 7.4 (30/70). Images of the fabricated hydrogels: (**b**) plain (not loaded with bilosomal dispersion), (**c**) medicated (loaded with bilosomal dispersion), and (**d**) diagram depicting the formation of an interpenetrating 3D network of curcumin bilosomal hydrogel using calcium cross-linkages.

**Figure 4 pharmaceutics-16-00090-f004:**
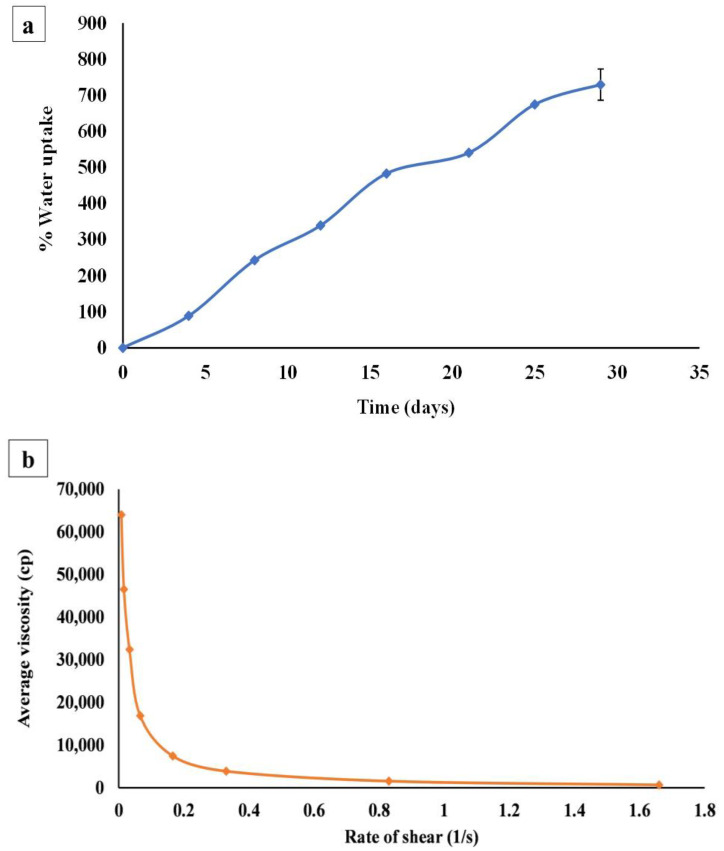
(**a**) Mean water uptake capacity of curcumin bilosomal (F5) hydrogel placed in phosphate-buffered saline at room temperature (25 °C) over four weeks (*n* = 3), (**b**) mean rheological properties of bilosomal hydrogel (*n* = 3), (**c**) TEM micrographs of free bilosomal formulation F5 (scale bar = 0.5 µm), (**d**) TEM micrographs of bilosomal hydrogel (scale bar = 1 µm).

**Figure 5 pharmaceutics-16-00090-f005:**
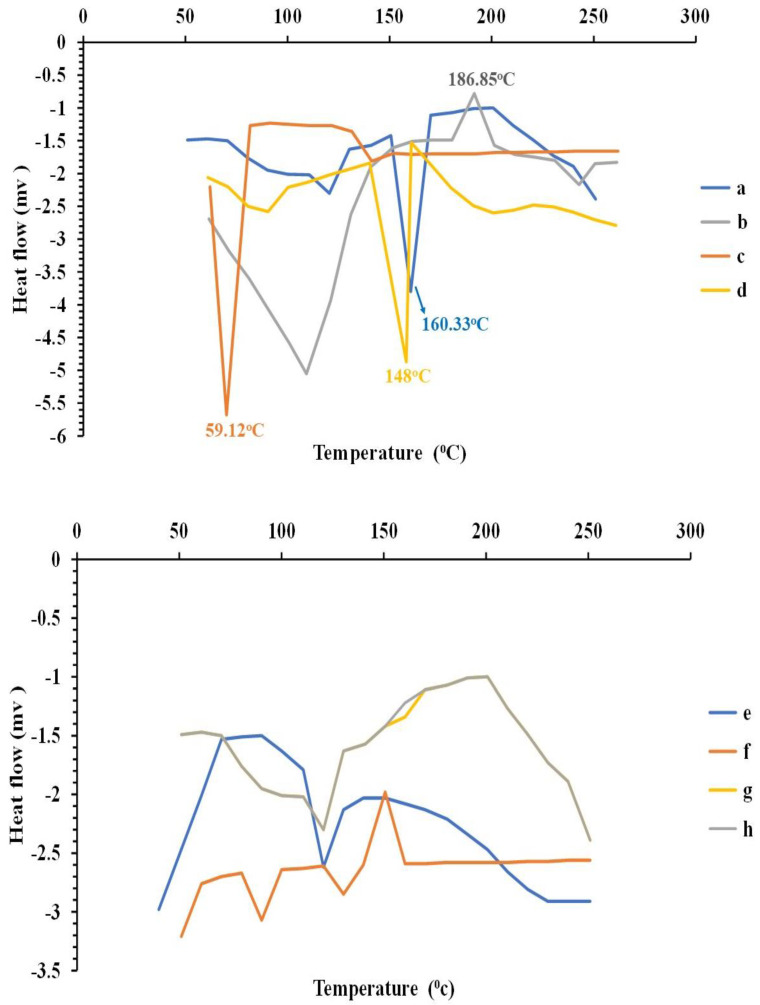
DSC thermograms of (a) curcumin, (b) sodium deoxycholate, (c) Span^®^ 60, (d) cholesterol, (e) free bilosomal formulation F5, (f) bilosomal formulation F5 physical mixture, (g) curcumin-loaded bilosomal hydrogel, and (h) plain hydrogel.

**Figure 6 pharmaceutics-16-00090-f006:**
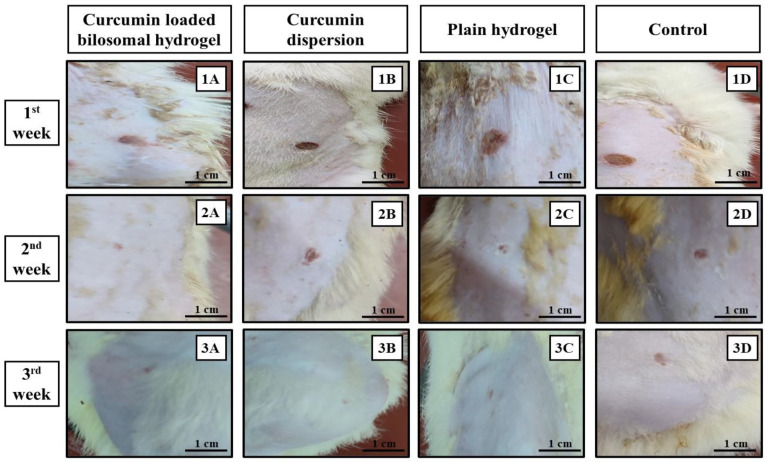
Images for the wound healing process during the three-week treatment period. (**1A**,**2A**,**3A**) Curcumin-loaded bilosomal hydrogel, (**1B**,**2B**,**3B**) curcumin dispersion, (**1C**,**2C**,**3C**) plain hydrogel, and (**1D**,**2D**,**3D**) control. (Scale bar = 1 cm).

**Figure 7 pharmaceutics-16-00090-f007:**
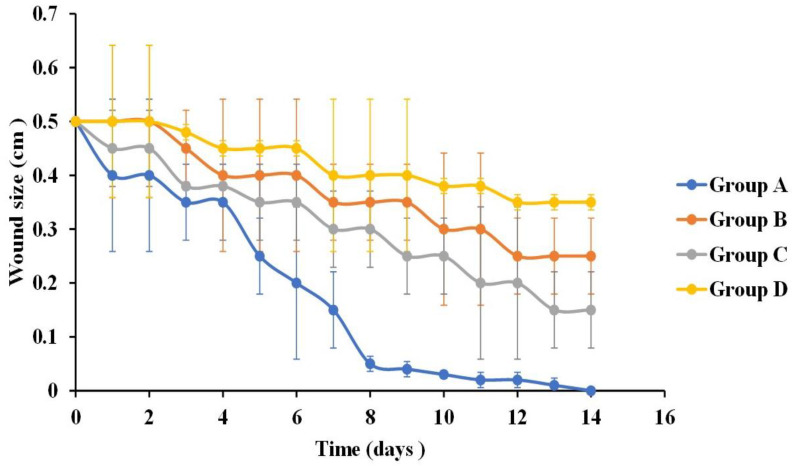
Mean wound size change after 14 days of the application of curcumin-loaded bilosomal hydrogel (group A), curcumin dispersion (group B), plain hydrogel (group C), and control (group D) (*n* = 3), Error bars represent standard deviations.

**Figure 8 pharmaceutics-16-00090-f008:**
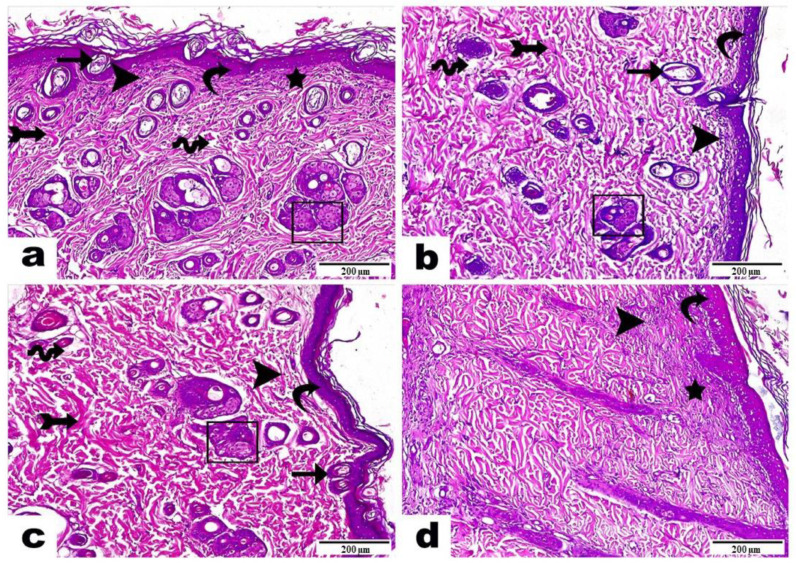
Histopathological examination of wounds in rat skin samples three weeks post treatment with (**a**) curcumin-loaded bilosomal hydrogel, (**b**) curcumin dispersion, (**c**) plain hydrogel, and (**d**) untreated control. Hematoxylin & Eosin stains were used. Magnification power = 100×, scale bar = 200 μm. Curvy arrow: epidermis; arrow: migrating epidermal cells; rectangle: skin gland development; arrow with tail: dermis layer; star: thin fibers; wavy arrow: interstitial edema; arrowhead: inflammatory cell infiltrations.

**Table 1 pharmaceutics-16-00090-t001:** Composition and characterization of curcumin-loaded bilosomal formulations and bilosomal hydrogel.

Formulation Code	Type (Factor A) and Amount (Factor B) of Bile Salts	Cholesterol/Span^®^ 60 Molar Ratio * (*w*/*w*) (Factor C)	PS (nm)	PDI	EE% (%)	Drug Content (%)	ZP (mV)	Q_2h_ (%)	Q_72h_ (%)
F1	SDC (5 mg)	1:5	197.10 ± 24.60	0.460 ± 0.040	96.59 ± 0.10	97.80 ± 0.14	−32.10 ± 0.21	12.68 ± 0.14	28.48 ± 0.10
F2	SC (5 mg)	1:5	448.00 ± 15.40	0.490 ± 0.030	89.50 ± 0.98	85.15 ± 0.21	−29.20 ± 0.84	15.66 ± 0.14	53.78 ± 1.00
F3	SDC (10 mg)	1:5	363.05 ± 8.55	0.474 ± 0.010	92.50 ± 2.20	92.80 ± 0.14	−38.90 ± 0.28	29.44 ± 0.14	42.40 ± 1.40
F4	SC (10 mg)	1:5	513.40 ± 21.80	0.454 ± 0.020	97.75 ± 1.50	89.10 ± 0.14	−34.20 ± 0.14	13.76± 0.70	41.25 ± 0.50
F5	SDC (5 mg)	1:10	246.25 ± 11.85	0.339 ± 0.030	93.32 ± 0.40	95.12 ± 0.03	−36.75 ± 0.14	38.42 ± 0.14	96.23 ± 0.02
F6	SC (5 mg)	1:10	627.80 ± 42.50	0.537 ± 0.050	86.60 ± 0.70	98.20 ± 0.28	−28.60 ± 0.56	11.19 ± 0.70	29.83 ± 0.30
F7	SDC (10 mg)	1:10	402.10 ± 4.00	0.650 ± 0.070	85.67 ± 0.34	90.10 ± 0.14	−38.80 ± 0.14	19.52 ± 0.14	40.07 ± 0.20
F8	SC (10 mg)	1:10	680.45 ± 9.65	0.531 ± 0.150	88.06 ± 0.10	93.65 ± 0.91	−30.35 ± 0.14	14.31 ± 0.14	28.16 ± 0.03
Curcumin-loaded bilosomal hydrogel	SDC (5 mg)	1:10	212.00 ± 25.03	0.434 ± 0.050	90.55 ± 0.21	91.72 ± 0.16	−54.00 ± 2.82	14.55 ± 0.14	74.08 ± 0.70

Data presented as the mean values of trials (*n* = 3) ± SD. Abbreviations. SDC: sodium deoxy cholate, SC: sodium cholate, PS: particle size, PDI: polydispersity index, ZP: zeta potential, EE%: entrapment efficiency %, Q_2h_: percentage of drug released after 2 h, Q_72h_: percentage of drug released after 72 h. * All cholesterol/Span^®^ 60 ratios were calculated based on *w*/*w* molar ratio.

## Data Availability

The data presented in this study are available in the article and Appendix A.

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
