# Peer review of "Cross-Linked Alginate Dialdehyde/Chitosan Hydrogel Encompassing Curcumin-Loaded Bilosomes for Enhanced Wound Healing Activity"

_pharmaceutics, 2024, doi:10.3390/pharmaceutics16010090_

Round 1
Reviewer 1 Report (New Reviewer)
Comments and Suggestions for Authors
In my opinion the manuscript is too long. It contains many results which is good, but the story that connects everything is missing, therefore this makes the manuscript somewhat hard to follow. Also, the Conclusion could be better written to integrate all results.
Abstract:
Polydispersity index is not in units [nm].
What is "particle size", is this Z-average? This needs to be clarified and used throughout the manuscript.
The decimal places in errors should correspond to the decimal places in measured values (this comment is applicable to the entire manuscript).
Clarify "besides it showed sustained drug release for five days (100.030%±0.007)". Does this mean 100% drug was released after five days. Please don't use decimal places here.
Section 2.3.2.
It is necessary to clarify the first sentence.
Section 2.6.3.
Correct "withdrawing"
Section 2.6.4.
Correct "dialysis bag"
It is not corret to write "under" Section 2.6.x
Figure 2 needs to be better quality.
3.4.7. Microscopy not microscope
3.4.8. Correct title
Comments on the Quality of English LanguageModerate editing of English language required
Author Response
Authors’ Response to Reviewer #1:
- In my opinion the manuscript is too long. It contains many results which is good, but the story that connects everything is missing, therefore this makes the manuscript somewhat hard to follow. Also, the Conclusion could be better written to integrate all results.
The comment is really appreciated. The manuscript was written based on the sequence of search attempts and results in the following order.
Firstly, sections (2.2) and (2.3) include fabrication of alginate dialdehyde (ADA) owing to its advantages over the native sodium alginate. Then, the characterization of the newly formed compound (ADA) such as determining the extent of oxidation of this reaction and Fourier transform infrared spectroscopy (FT-IR) to check for any new interactions, bonds or groups.
Secondly, section (2.4) includes the formation of ADA / chitosan hydrogel, where the hydrogel was regarded as an excellent formulation-carrier to be used in wound healing applications owing to its wide benefits in this field previously mentioned in manuscript. Several trails were carried to get the best consistency of hydrogel to be suitable for wound healing applications like gelation time test.
Thirdly, a nano system to improve the wound healing applications was prepared. So, sections (2.5) and (2.6) include the preparation and characterization of curcumin bilosomes.
Finally, sections (2.7) and (2.8) include the incorporation of curcumin bilosomes in the prepared hydrogel and the characterization of this newly formed bilosomal hydrogel and evaluating the healing ability of the hydrogel in-vivo on induced-wounds in rats.
The conclusion was modified to integrate all results as follows:
This study presented a simple technique for the fabrication of curcumin-loaded bilosomal hydrogel used in wound healing applications. Curcumin bilosomal hydrogels offered many advantages which made them efficient treatment for wound healing for various aspects. ADA was fabricated through oxidation of sodium alginate and then combined with chitosan to form a hydrogel with a gelation time 6.75 ± 0.353 min. Curcumin bilosomes were prepared and the best formulation composed of 5 mg SDC and Cholesterol / Span® 60 ratio (1:10) possessed PS of 246.25±11.85 nm, PDI of 0.339±0.030, ZP of -36.75±0.14 mV, drug content of 95.12±0.03, EE% of 93.32±0.40 and Q72h of 96.23±0.02 was incorporated in the prepared hydrogel.
The curcumin-loaded bilosomal hydrogel was able to sustain the curcumin release over five days. The hydrogel showed high % of water uptake at four-weeks’ time point (729.50 ± 43.13%) as well as shear-thinning behavior. The animal study showed that after three weeks of wound induction and treatment, the groups A, B and C showed completely closed wounds with a slight line scar. Wounds treated with only curcumin dispersion (group B) and hydrogel (group C) showed a slightly larger scar than those treated with curcumin bilosomal hydrogel (group A). On the contrary, control group D showed incomplete wound healing and closure with wound size (0.35 mm ± 0.014) and an obvious long-line scar. Histopathological studies proved the superior behavior of the curcumin-loaded bilosomal hydrogel in wound healing where the wounds were completely closed after a three-weeks’ period with programmed healing mechanism. This research presented the development of curcumin-loaded bilosomal alginate dialdehyde / chitosan hydrogel as an efficient, affordable, and promising wound treatment. Finally, the success of curcumin-loaded bilosomal hydrogel may be because of the synergistic action of curcumin-loaded bilosomes along with the cross-linked alginate dialdehyde / chitosan hydrogel.
- Abstract: Polydispersity index is not in units [nm].
The mistake is corrected and [nm] unit is deleted.
- What is "particle size", is this Z-average? This needs to be clarified and used throughout the manuscript.
The Z-average has been used to represent the particle size throughout the manuscript. The note has been added in the PS characterization section (2.6.1) in the manuscript.
- The decimal places in errors should correspond to the decimal places in measured values (this comment is applicable to the entire manuscript).
The comment has been followed. The decimal places in errors and measured values are modified to be the same.
- Clarify "besides it showed sustained drug release for five days (100.030%±0.007)". Does this mean 100% drug was released after five days. Please don't use decimal places here.
The percentage of drug released on the 5th day was 100%. The decimal had been removed.
- Section 2.3.2. It is necessary to clarify the first sentence.
“FT-IR spectroscopy was performed to check for the new interactions after formation of ADA and comparing it with pure NaAlg.”
The word “interactions” was replaced with “functional groups” in the manuscript to be more clear.
The oxidation reaction of alginate with sodium periodate is performed on hydroxyl groups at C-2 and C-3 positions of the uronates generating dialdehyde functional groups on the alginate backbone. Owing to their reactivity, these newly introduced aldehyde groups can interact with the hydroxyl groups present either on the adjacent uronic acid subunits in the same or adjacent polymer chains to form (intramolecular and intermolecular) hemiacetals, respectively, leading to a scarceness in the available aldehyde groups.
- Section 2.6.3.: Correct "withdrawing"
The word has been corrected.
- Section 2.6.4. Correct "dialysis bag" It is not correct to write "under" Section 2.6.x
The word has been changed to dialysis membrane.
We are representing the in vitro curcumin release as one of the different methods of characterization of curcumin bilosomes so it appears under the section of “2.6. Characterization of the prepared curcumin bilosomes”.
The same was done later under section “2.8. Characterization of the prepared curcumin bilosomal hydrogel”, where the in vitro release was presented as a test for characterization of the prepared curcumin bilosomal hydrogel.
- Figure 2 needs to be better quality.
The quality of the figure is improved.
- 4.7./ 3.4.8. Correct title (Microscopy not microscope)
The titles are corrected.
Comments on the Quality of English Language (Moderate editing of English language required)
English language revision was done.
Reviewer 2 Report (New Reviewer)
Comments and Suggestions for Authors
The manuscript entitled “Cross-linked alginate dialdehyde/chitosan hydrogel encompassing curcumin-loaded bilosomes for enhanced wound healing activity” by Sideek et al. demonstrated an alginate/chitosan system loaded with curcumin for wound healing application. As the authors know and stated in the introduction as well as formerly reviewed the studies in this area (ref3.) Sideek, S.A., et al. Pharmaceutics, 2023. 15, DOI: 10.3390/pharmaceutics15010038.), there are plenty of similar systems in the literature. However, this study distinguished itself from previous work by incorporating the liposomes of curcumin with alginate/chitosan system. The histopathology shows differences between control and the proposed system which is an indicator that the system is working. However, the presentation of data, some analyses and discussions need to be reconsidered/improved to make this article suitable for publication. Below, please kindly find my comments:
· Figure-2 FTIR spectra are hardly readable and thus not publishable.
· Section3.2.1. Gelation time. The gelation time should be defined here, e.g., what were the conditions that the author decided the polymer was gelled (e.g., temperature, …). The author later stated that “by examining the formed hydrogel, it was found that it was not firm enough and was in a state of a viscous liquid”. Conventionally, the gelation test is conducted through visual examination of the vial in a way that if the vial is physically placed in an angle (not straight) the material does not move (gelled). Was this the case for this experiment? If so, perhaps the inclusion of photos of these would help.
· Table 1. Can the authors elaborate, clarify, or confirm if the digits after the decimal points are actually the accuracy of the measurement (measuring equipment)?
· Transmission electron microscopy (TEM): As the author stated, “A sample drop was placed on a carbon-coated copper grid surface and left to dry for 1 min (for F5 dispersion) and 5 min (for the hydrogel); any excess was dried by a tip of filter paper.” The author later stated in section 3.4.7: “Photomicrographical images of the two examined samples; free bilo-somes (F5) and bilosomal hydrogel, showed non-aggregating spherical vesicles with well-defined walls and smooth surfaces as shown in Figure (5c & 5d
· respectively). This was a good indication that the incorporation of the optimal curcumin-loaded bilosomes formulation (F5) in the hydrogel did not affect their shape.”
I understand that imaging such as SEM or TEM have been conducted in the literature, unfortunately. However, as the authors appreciate, when a hydrogel is dried, it becomes dehydrated which is leading to deswelling. Subsequently, the pores of dried gel networks are much smaller than in actual gel networks. Thus, it is hard for me to infer the same conclusion as the authors merely by Figure 5 c-d.
· Section 3.4.8. “DSC thermogram of curcumin showed a sharp endothermic peak at160.33⁰C which indicated the melting point of its crystalline form [41, 97]. Span® 60 DSC thermogram revealed an exothermic peak at 59.12⁰C representing its transition temperature [98, 99]. SDC gave an endothermic peak at 186.85⁰C [100]. Cholesterol exhibited an endothermic peak indicating its melting point at 148oC [101, 102].” Can the author label these peaks on the spectra? It is hard to analyse these in this form. Also, can the authors confirm that they perform heating and cooling cycles on all these DSCs and if not, what was the rationale behind not doing so?
· Figure 7. The scale at which these images are taken makes it hard to compare them against each other, e.g., the distance of the camera to the wound in 3A, 3B, and 3C (in particular) is different than that of “control” (3D).
· Figure 8. What are the error bars here? It is odd that the error bars (at least in the way that they appear to me) of each graph within themselves are the same. Consider double-checking the statistics here and mentioning what the error bars are here on this graph.
Comments on the Quality of English LanguageOverall, the quality of English is ok, however, I am personally not a huge admirer of using superlative/ qualitative adjectives. Example: using "marvellous" for curcumin can be reconsidered.
Author Response
Authors’ Response to Reviewer #2:
- Figure-2 FTIR spectra are hardly readable and thus not publishable.
The quality of the figure is improved.
- Section 3.2.1. Gelation time. The gelation time should be defined here, e.g., what were the conditions that the author decided the polymer was gelled (e.g., temperature, …). The author later stated that “by examining the formed hydrogel, it was found that it was not firm enough and was in a state of a viscous liquid”. Conventionally, the gelation test is conducted through visual examination of the vial in a way that if the vial is physically placed in an angle (not straight) the material does not move (gelled). Was this the case for this experiment? If so, perhaps the inclusion of photos of these would help.
The methodology and conditions for determining the gelation time was mentioned under section:
2.4.2. Determination of gelation time
The gelation time test was performed according to a previously report-ed method with minor modifications [39]. A mixture of CS and ADA solution treated with CaCl2 solution was transferred to a Petri dish (100 × 20 mm2), and a magnetic stirring bar (Teflon fluorocarbon resin, 5 × 2 mm2) was placed in the center and the solution was stirred at 300 rpm using a hot plate/stirrer adjusted at 40 °C. The gelation time was recorded when the solution formed a solid globule that interfered with the magnetic stirrer bar and separated completely from the bottom of the dish [15].
So we relied on physical examination to check for gel formation and a photo in Figure (3b) is included to show the formed plain hydrogel.
The sentence “by examining the formed hydrogel, it was found that it was not firm enough and was in a state of a viscous liquid” meant that the gel formed without a crosslinker had a soft texture.
The sentence was modified in the manuscript to be clearer and “was in a state of a viscous liquid” was deleted.
- Table 1. Can the authors elaborate, clarify, or confirm if the digits after the decimal points are actually the accuracy of the measurement (measuring equipment)?
The digits after the decimal points are selected as per the measuring equipment.
- Transmission electron microscopy (TEM): As the author stated, “A sample drop was placed on a carbon-coated copper grid surface and left to dry for 1 min (for F5 dispersion) and 5 min (for the hydrogel); any excess was dried by a tip of filter paper.” The author later stated in section 3.4.7: “Photomicrographical images of the two examined samples; free bilosomes (F5) and bilosomal hydrogel, showed non-aggregating spherical vesicles with well-defined walls and smooth surfaces as shown in Figure (5c & 5d respectively). This was a good indication that the incorporation of the optimal curcumin-loaded bilosomes formulation (F5) in the hydrogel did not affect their shape.”
I understand that imaging such as SEM or TEM have been conducted in the literature, unfortunately. However, as the authors appreciate, when a hydrogel is dried, it becomes dehydrated which is leading to deswelling. Subsequently, the pores of dried gel networks are much smaller than in actual gel networks. Thus, it is hard for me to infer the same conclusion as the authors merely by Figure 5 c-d.
The reviewer’s comment is appreciated. The aim of TEM imaging is to see the changes in shape and size of curcumin bilosomes. So, even if the pores of dried gel networks became much smaller, this had not affected the shape and size of the incorporated bilosomes.
- Section 3.4.8. “DSC thermogram of curcumin showed a sharp endothermic peak at160.33⁰C which indicated the melting point of its crystalline form [41, 97]. Span® 60 DSC thermogram revealed an exothermic peak at 59.12⁰C representing its transition temperature [98, 99]. SDC gave an endothermic peak at 186.85⁰C [100]. Cholesterol exhibited an endothermic peak indicating its melting point at 148oC [101, 102].” Can the author label these peaks on the spectra? It is hard to analyse these in this form. Also, can the authors confirm that they perform heating and cooling cycles on all these DSCs and if not, what was the rationale behind not doing so?
The peaks are labelled on the spectra and the figure was modified. The DSC was done to determine the exact melting points of the reagents and compare them with the reported data to ensure their purity before using them in the formulation. Therefore, only heating cycle was needed to determine the melting point of each compound and indeed they were consistent with the reported data.
- Figure 7. The scale at which these images are taken makes it hard to compare them against each other, e.g., the distance of the camera to the wound in 3A, 3B, and 3C (in particular) is different than that of “control” (3D).
The authors tried to take pictures of the wounds from the same angle and distance as possible. The difference of camera distance in different images might be due to bad cropping of the images to show only one wound per image.
Modifications to the images had been made to make the scale more comparable and the figure had been replaced.
- Figure 8. What are the error bars here? It is odd that the error bars (at least in the way that they appear to me) of each graph within themselves are the same. Consider double-checking the statistics here and mentioning what the error bars are here on this graph.
The error bars represent the standard error between the different measurements of wound sizes at each time in each group.
The authors appreciate the comment. The statistics had been checked and a mistake in error bars drawing had been found, making them the same for each graph. The error bars had been corrected and the figure was replaced.
Comments on the Quality of English Language
Overall, the quality of English is ok, however, I am personally not a huge admirer of using superlative/ qualitative adjectives. Example: using "marvellous" for curcumin can be reconsidered.
This comment was appreciated and the whole manuscript was revised for qualitative adjectives and they were exchanged for others.
Reviewer 3 Report (New Reviewer)
Comments and Suggestions for Authors
The authors have made an interesting article describing the formation and characterization of cross-linked alginate dialdehyde/chitosan hydrogel encompassing curcumin-loaded bilosomes for enhanced wound healing activity but some improvements should be made:
1. In the abstract it is written “In vivo animal testing and histopathological studies supported the superiority of the curcumin-loaded bilosomal hydrogel in wound healing where there was a complete wound closure attained after three weeks period with proper healing mechanism” it should be specified the superiority in comparison with what?
2. In the introduction, regarding the hydrogels in the wound healing it should include the smart hydrogel, where it can be considered citing newer paper such as: Pharmaceutics | Free Full-Text | New Smart Bioactive and Biomimetic Chitosan-Based Hydrogels for Wounds Care Management (mdpi.com)
3. In the introduction the phrase” as it comprises the summation of both benefits and magnification of their beneficial effects” should be rephrased because it repeats the benefits/beneficial.
4. In the introduction “in-vitro release and in-vivo” in vitro and in vivo should be written with italic
5. In the Material and Methods “phyto chem sciences Inc” should be written with capital letters
6. At Section 2.3.1.-Determination of oxidation degree” the periodate concentration in the sample was deduced using the molar absorption coefficient previously calculated from absorbance of the complex versus iodate concentration (0.800–2.105 M)” should be elaborated. How it was deduced? What formula was applied?
7. At Section 2.3.1.-Determination of oxidation degree” In the formula -Oxidation degree = (consumed amount of periodate / added amount of periodate) * 100, the consumed amount of periodate expressed in what? Molar concentration, percentage, mg?
8. For the drug content, “preconstructed calibration curve in PBS ethanol / phosphate buffer saline” was performed using how many concentration and which concentrations?
9. For the calculation of EE % = (Amount of entrapped drug/Actual drug content) × 100, the amount of entrapped drug expresses as…..???
10. How the in vitro curcumin release from bilosomes was calculated?
11. In Section 2.7. Fabrication of curcumin bilosomal hydrogel, whate were the concetrations of CaCl2 solutions?
12. Pleasse check the uniformity of writing in vitro, in vivo with italic everywhere.
13. In 2.10. In vivo animal study 2.10.1. Animal- how 6 rats were distributed in 4 groups?
14. In the animal model-“four full thickness rounded skin excision wounds of 5 mm in diameter were performed on both sides of the spine of each single rat using a sterile biopsy punch needle” so each rat had 8 rounded excision wound (4 on each side of the spine). From the Figure 7 it is not visible all the 8 wounds.
15. In section 2.10.2 reformulate the phrase “the rats were mercifully terminated”
16. Figure 2 should be redone because the scale with numbers it is not visible, and also it will be better a superposed spectra with all three spectra.
17. In Table 1, SDC it is not defined
18. I consider that the Figure can be regrouped so it will take up less space.
Author Response
Authors’ Response to Reviewer #3:
- In the abstract it is written “In vivo animal testing and histopathological studies supported the superiority of the curcumin-loaded bilosomal hydrogel in wound healing where there was a complete wound closure attained after three weeks period with proper healing mechanism” it should be specified the superiority in comparison with what?
The superiority of the curcumin-loaded bilosomal hydrogel in wound healing was indicated when compared to curcumin dispersion and plain hydrogel as mentioned in the in-vivo study section.
This clarification was also added to the abstract.
- In the introduction, regarding the hydrogels in the wound healing it should include the smart hydrogel, where it can be considered citing newer paper such as: Pharmaceutics | Free Full-Text | New Smart Bioactive and Biomimetic Chitosan-Based Hydrogels for Wounds Care Management (mdpi.com).
A new paragraph was added in the introduction which briefly explain the smart hydrogels with citing some new papers.
- In the introduction the phrase” as it comprises the summation of both benefits and magnification of their beneficial effects” should be rephrased because it repeats the benefits/beneficial.
The phrase has been rephrased into “as it comprises the summation of both benefits and amplifies their positive effects”.
- In the introduction “in-vitro release and in-vivo” in vitro and in vivo should be written with italic.
“In vitro” and “in vivo” has been changed to italic in all the manuscript.
- In the Material and Methods “phyto chem sciences Inc” should be written with capital letters.
The correction has been made.
- At Section 2.3.1.-Determination of oxidation degree” the periodate concentration in the sample was deduced using the molar absorption coefficient previously calculated from absorbance of the complex versus iodate concentration (0.800–2.105 M)” should be elaborated. How it was deduced? What formula was applied?
The exact concentration used in the procedures was adjusted to be (0.082 M/L).
Section 2.3.1 described the equation of oxidation degree as following:
The oxidation percentage was determined using the following equation (using absorbance):
Oxidation degree = (absorbance of consumed amount of periodate / absorbance added amount of periodate) * 100.
The detailed calculations are provided here as following:
For the blank test: (without NaAlg)
(the mix solution of KI, starch in phosphate buffer saline, sodium periodate in ethanol and distilled water)
Absorbance = 3.032 (1)
Knowing that (1) is the reaction blank
For the reaction of NaAlg with periodate:
(the mix solution of KI, starch in phosphate buffer saline, NaAlg, sodium periodate in ethanol and distilled water)
Absorbance =
0.678 (2)
(1)-(2) =2.354 , which resemble the consumed quantity of sodium periodate
Oxidation degree = (absorbance of consumed amount of periodate / absorbance of added amount of periodate ) * 100 = (2.354 / 3.032 ) * 100 = 77.6 %
The oxidation percentage was determined using the following equation (using concentration):
For the blank test: (without NaAlg)
(the mix solution of KI, starch in phosphate buffer saline, sodium periodate in ethanol and distilled water)
Absorbance = 3.032
Knowing that (1) is the reaction blank
Since the concentration of used sodium periodate was in gm (1.76 g in 100 ml), we had to convert it to Mol/L to cope with the molar absorptivity as following:
(1.76 /0.1*213.89) = 0.082 Mol/L (1)
Molar absorptivity = (Absorbance)/ (concentration *path length) (Mol/L/cm) = 3.032/(0.082*1) = 36.97 Mol/L/cm
For the reaction of NaAlg with periodate:
(the mix solution of KI, starch in phosphate buffer saline, NaAlg, sodium periodate in ethanol and distilled water)
Absorbance = 0.678
Absorbance = molar absorptivity * concentration
0.678 = 36.97 * concentration
Concentration = 0.0183 Mol/L (2)
Oxidation degree = (concentration of consumed amount of periodate / concentration of added amount of periodate) * 100 = (0.082 – 0.0183)/(0.082) *100 = 77.6%
- At Section 2.3.1.-Determination of oxidation degree” In the formula -Oxidation degree = (consumed amount of periodate / added amount of periodate) * 100, the consumed amount of periodate expressed in what? Molar concentration, percentage, mg?
The consumed amount was expressed in Mol/L. The calculations was provided in the previous comment.
- For the drug content, “preconstructed calibration curve in PBS ethanol / phosphate buffer saline” was performed using how many concentration and which concentrations?
The calibration curve was constructed by measuring the absorbance of curcumin solution in concentration range 2-20 µg/mL. Serial dilutions of 10 concentrations from 2-20 µg/mL were prepared.
The measured concentrations were 2, 4, 6, 8, 10, 12, 14, 16, 18, and 20 µg/mL.
- For the calculation of EE % = (Amount of entrapped drug/Actual drug content) × 100, the amount of entrapped drug expresses as…..???
The amount of entrapped drug was expressed as mg.
- How the in vitro curcumin release from bilosomes was calculated?
As stated in section 2.6.4.
Certain volumes of bilosomes (equivalent to 2 mg curcumin) in addition to 1.5 mL of release medium were transferred to a dialysis cellulose membrane. Each bag was transferred to an amber colored glass light-resistant bottle (wrapped with aluminum foil) containing 100 mL of release medium. The glass bottles were placed in a thermo-statically controlled operating at 100 rpm and maintained at 37 ± 0.2 °C. At definite time intervals (1, 2, 3, 4, 6, 8, 24, 48, 72, 96 and 120 h), aliquots of 3 mL were withdrawn The withdrawn samples were measured spectrophotometrically at 424 nm.
% Curcumin released = (A x K x volume of release medium x 100) / (1000 x dose of dug)
Where A is absorbance and K is inverse of slope of calibration curve.
- In Section 7. Fabrication of curcumin bilosomal hydrogel, what were the concentrations of CaCl2 solutions?
A 1% CaCl2 solution was used. This is added to the manuscript under section 2.7.
- Please check the uniformity of writing in vitro, in vivo with italic everywhere.
“In vitro” and “in vivo” has been changed to italic in all the manuscript.
- In 2.10. In vivo animal study 2.10.1. Animal- how 6 rats were distributed in 4 groups?
Thank you for pointing this out. It was mistakenly typed. Eight rats were used and distributed into 4 groups. The mistake is corrected throughout the manuscript.
- In the animal model-“four full thickness rounded skin excision wounds of 5 mm in diameter were performed on both sides of the spine of each single rat using a sterile biopsy punch needle” so each rat had 8 rounded excision wound (4 on each side of the spine). From the Figure 7 it is not visible all the 8 wounds.
Each rat has four rounded excision wounds (2 on each side of the spine). The phrase in the manuscript was changed to “four full thickness rounded skin excision wounds of 5 mm in diameter were performed, two on each side of the spine of each single rat” to be clearer.
Only one wound of the four was photographed as a representation of the wound healing effect of the treatment.
- In section 2.10.3 reformulate the phrase “the rats were mercifully terminated”
The phrase had been reformulated to “the rats were euthanized”.
- Figure 2 should be redone because the scale with numbers it is not visible, and also it will be better a superposed spectra with all three spectra.
The quality of the figure has been improved.
The FTIR spectra are provided to us as separate images for each component, so it is difficult to make a superposed spectra with all the three spectra.
- In Table 1, SDC it is not defined
The definition of SDC had been added to the footnote of Table 1.
- I consider that the Figure can be regrouped so it will take up less space.
The authors appreciate the reviewer’s comment. We grouped some figures in the manuscript. But any further regrouping will affect the quality of the figures and the flow of the manuscript.
Round 2
Reviewer 2 Report (New Reviewer)
Comments and Suggestions for Authors
The authors addressed my comments. Below, please see my minor comments:
1) Given the changes in errors of Figure 7: They should mention in the caption of Figure 7 what the error bars are. Also, they may need to conduct appropriate statistical analysis on this data for comparison.
2) The justification for doing only the heating cycle of DSC should be added to the manuscript main text.
Author Response
Authors’ Response to Reviewer #2:
- Given the changes in errors of Figure 7: They should mention in the caption of Figure 7 what the error bars are. Also, they may need to conduct appropriate statistical analysis on this data for comparison.
Error bars represent standard deviations and the caption of figure 7 was highlighted purple and modified as following:
Figure 7. Mean wound size change after 14 days of the application of curcumin-loaded bilosomal hydrogel (group A), curcumin dispersion (group B), plain hydrogel (group C), and control (group D) (n = 3), Error bars represent standard deviations.
Statistical analysis of wound closure was mentioned in section 3.5.1(highlighted purple)
3.5.1. Rate of wound closure
Statistical analysis of the wound size reduction after one-week post-treatment revealed that the skin wounds in groups A, B, and C showed a significantly smaller wound size (0.15 ± 0.002 cm, 0.35 ± 0.007 cm, and 0.3 ± 0.014 cm, respectively) compared to the control group D (0.4 ± 0.014 cm) (p ˂ 0.05) (Figure 7). In addition, comparing the groups, A, B, and C with each other revealed that group A was statistically significant (p ˂ 0.01) than groups (B and C), whereas groups (B and C) showed non-statistical significance from each other (p = 0.09).
- The justification for doing only the heating cycle of DSC should be added to the manuscript main text.
The justification was highlighted purple and added to (Discussion of DSC) section 3.4.8.
3.4.8. Differential scanning calorimetry (DSC)
The DSC was done to determine the exact melting points of the used reagents and compare them with the reported data to ensure their purity before using them in the formulation. Therefore, only heating cycle was needed to determine the melting point of each compound.
Reviewer 3 Report (New Reviewer)
Comments and Suggestions for Authors
The authors made the proper modifications according to the requirements from the comments, so the article is ready for publication.
Author Response
Authors' response to Reviewer #3 comments:
The authors made the proper modifications according to the requirements from the comments, so the article is ready for publication.
Thanks.
This manuscript is a resubmission of an earlier submission. The following is a list of the peer review reports and author responses from that submission.
Round 1
Reviewer 1 Report
Comments and Suggestions for Authors
Authors fabricated a curcumin-loaded hydrogel via thin film hydration technique using cholesterol for topical wound healing. Bilosomes were optimized by PS, PDI, ZP and EE% properties. And the loaded hydrogel presented superiority in wound healing after three weeks. But the paper is of the state of unfinished work and there are some questions need be promoted before the next.
1. Bar size should be marked in all photos, in Figure 3 & 5. And the size should be micrometer (μm) not UM in Figure 7.
2. In 2.9, authors said statistical analysis has been done using PSSS@ software, but there did not present the statistical results shown in all figures.
3. Authors performed the animal model on Albino rats with a full thickness rounded skin excision wounds of 5 mm in diameter. As we known, the critical defect of rat is 8 mm -12 mm for the skin repairing. How did authors think about such a defect model. Similar researches have been presented in such a field as following:
[1] Meng Li, Jing Chen, Mengting Shi, Hualei Zhang, Peter X. Ma, Baolin Guo*, Electroactive anti-oxidant polyurethane elastomers with shape memory property as non-adherent wound dressing to enhance wound healing, Chemical Engineering Journal, 375(2019) 121999
[2] Nandin Mandakhbayar, YunSeong Ji, Ahmed El-Fiqi, Double hits with bioactive nanozyme based on cobalt-doped nanoglass for acute and diabetic wound therapies through anti-inflammatory and pro-angiogenic functions, Bioactive Materials 31 (2024) 298–311
[3] Mohammad Hadi Norahan, Sara Cristina Pedroza-Gonz′alez, M′onica Gabriela S′anchez-Salazar, Mario Mois′es ′Alvarez, Grissel Trujillo de Santiago, Structural and biological engineering of 3D hydrogels for wound healing, Structural and biological engineering of 3D hydrogels for wound healing, Bioactive Materials 24 (2023) 197–235
4. In 2.8.8, “a sample drop was placed on a carbon-coated copper grid surface and left to dry for 1 min”. How is about the dried condition for the hydrogel beads?
5. In 2.10. In vivo animal study, how is about the sterilized condition for tested samples?
6. Please plot the FTIR spectra in one figure of sodium alginate, alginate dialdehyde and chitosan to compare the shift of these curves in Figure S1. And mark the specific peaks for characteristic groups.
7. In Table 1, what is the prepared condition for last sample (Curcumin loaded bilosomal hydrogel)? What is the mean EE% for the last sample?
8. Please mark the factors A, B, C for which formulation code group in Table 1. It is lost in these data with the description in 3.3.1.
9. In 3.3.1 Determination of PS, PDI & ZP, it is better to plot a figure to discuss the significant difference with the data shown in Table1. It is confused in the description with different conditions of experiment groups.
10. In Figure 5, only one defect has been observed in photos, but in 2.10.2, “four full thickness rounded skin excision wounds of 5 mm in diameter were performed on both side of the spine of each single rat”.
11. Figure 7, what is the meaning of the different arrows?
12. References are over-documented, it is better to reduce half.
Comments on the Quality of English Language
The logic and scientific expression need be improved for understanding.
Reviewer 2 Report
Comments and Suggestions for Authors
This is a very interesting article. The research methodology, conducting the experiment and processing experimental data are described in great detail. This does not raise doubts about the reliability of the obtained results. A few comments relate more to the design of the article rather than its content.
In paragraph 2.1. all materials are named, but none of their characteristics are given. For example, Chitosan, Span® 60, Xylazine, xyla-ject and other materials need clarification of their characteristics.
It is necessary to redo table 1. The word mean is repeated very often in the name of the columns.
Figure 5 is better to available online, but Figures S1, S2 are given in the text of the article.
Reviewer 3 Report
Comments and Suggestions for Authors
Please find my comments on your manuscript “Cross-linked alginate dialdehyde/chitosan hydrogel encompassing curcumin-loaded bilosomes for enhanced
wound healing activity”
- Considering synthetic alginate dialdehyde (ADA), Please bring this aim in “Introduction” section.
- In “Formation of ADA /chitosan hydrogel”, please mention hardening time.
- Please explain what is the privilege of “oxidized sodium alginate” to “Sodium alginate”?
- Figure 4a,b and Figure 5 have poor quality and resolution. Please improve it and make sure that all the labels and legends are clear and readable.
- Please add figure markup in the figure description.
- Considering that curcumin has been extensively studied, please refer to the most relevant and recent studies that are similar to your work and highlight the novelty and significance of your work. How does your work differ from or improve upon the existing literature?
Regards,
Reviewer 4 Report
Comments and Suggestions for Authors
The ms. is about cross-linked alginate dialdehyde/chitosan hydrogel encompassing curcumin-loaded bilosomes for enhanced wound healing activity. I can't suggest it for publication for the following points as below.
1. The information about the molecular weight and polydispersity of alginate should be added.
2. It can't be believed that the drug loading content is so high (around 90%).
3. The PDI of the bilosomes is too high for a well-dispersed system.
4. The quality of the figures is poor.
Comments on the Quality of English LanguageMinor editing of English language required